# LESS IS MORE: DIMENSION REDUCTION FINDS ON-MANIFOLD ADVERSARIAL EXAMPLES IN HARD-LABEL ATTACKS

## ABSTRACT

Designing deep networks robust to adversarial examples remains an open problem. Likewise, recent zeroth-order hard-label attacks on image classification models have shown comparable performance to their first-order, gradient-level alternatives. It was recently shown in the gradient-level setting that regular adversarial examples leave the data manifold, while their on-manifold counterparts are in fact generalization errors. In this paper, we argue that query efficiency in the zeroth-order setting is connected to an adversary's traversal through the data manifold. To explain this behavior, we propose an information-theoretic argument based on a *noisy manifold distance oracle*, which leaks manifold information through the adversary's gradient estimate. Through numerical experiments of manifold-gradient mutual information, we show this behavior acts as a function of the effective problem dimensionality. On high-dimensional real-world datasets and multiple zeroth-order attacks using dimension reduction, we observe the same behavior to produce samples closer to the data manifold. This can result in up to 4x decrease in the manifold distance measure, regardless of the model robustness. Our results suggest that taking the manifold-gradient mutual information into account can thus inform better robust model design in the future, and avoid leakage of the sensitive data manifold information.

## 1 INTRODUCTION

Adversarial examples against deep learning models were originally investigated as blind spots in classification (Szegedy et al., 2013; Goodfellow et al., 2014). Formal methods for discovering these blind spots emerged, which we denote as gradient-level attacks, and became the first techniques to reach widespread attention within the deep learning community (Papernot et al., 2016; Moosavi-Dezfooli et al., 2015; Carlini & Wagner, 2016; 2017; Chen et al., 2018). In order to compute the necessary gradient information, such techniques required access to the model parameters and a sizeable query budget. These shortcomings were addressed by the creation of score-level attacks, which only require the confidence values output by the deep learning models (Fredrikson et al., 2015; Tramèr et al., 2016; Chen et al., 2017; Ilyas et al., 2018). However, these attacks still rely on models to divulge information that would be impractical to receive in real-world systems. By contrast, hard-label attacks make no assumptions about receiving side information, and only the predicted class is observable, thus providing the weakest, yet most realistic adversarial threat model. These methods, which originated from a random-walk on the decision boundary (Brendel et al., 2017), have been carefully refined to offer convergence guarantees (Cheng et al., 2019), query efficiency (Chen et al., 2019; Cheng et al., 2020), and capability in the physical world Feng et al. (2020).

Despite the steady improvements of hard-label attacks, open questions persist about their behavior, and adversarial machine learning (AML) attacks at large. Adversarial examples were originally assumed to lie in rare pockets of the input space (Goodfellow et al., 2014), but this conventional wisdom was later challenged by the boundary tilting assumption (Tanay & Griffin, 2016; Gilmer et al., 2018), which adopts a "data-geometric" view of the input space living on a lower-dimensional manifold. This is supported by Stutz et al. (2019), who suggest that regular adversarial examples leave the data manifold, while on-manifold adversarial examples are generalization errors. From a data-geometric perspective, an adversarial example's distance to the manifold primarily describes

the amount of semantic features preserved during the attack process. This makes it advantageous to produce on-manifold adversarial examples, since the adversary can exploit the inherent generalization error of the model while producing samples that are semantically similar for humans. However, the *true* data manifold is either difficult or impossible to describe, and relying solely on approximations of the manifold can lead to the creation of crude adversarial examples (Stutz et al., 2019).

In this paper, we adopt the boundary-tilting assumption and demonstrate an unexpected benefit of query-efficient zeroth-order attacks, i.e., attacks enabled by the use of dimensionality reduction techniques. These attacks are more likely to discover on-manifold examples, which we theoretically demonstrate is the result of manifold-gradient mutual information. Our results suggest that this quantity can *increase* as a function of the data dimensionality. This information leakage leads to adversarial examples that are on-manifold generalization errors. With this knowledge, we empirically demonstrate how to improve hard-label attacks in a generic yet principled way, and potentially re-think their interaction with model robustness and public-facing systems in the near future.

For clarity, we provide a block diagram of our claims and experiments in the Appendix (Section A.3). Our specific contributions are as follows:

- **Introduction of manifold distance oracle.** To create on-manifold examples, the adversary must (implicitly) leverage manifold information during the attack phase. We thus propose an information-theoretic formulation of the noisy manifold distance (NMD) oracle, which can explain how zeroth-order attacks craft on-manifold examples. We theoretically demonstrate on a Gaussian data model that *manifold-gradient mutual information can increase as a function of data dimensionality*. We empirically show this is true even on large-scale image datasets such as CIFAR-10 and ImageNet. This finding relates to known behavior in the gradient-level setting, where semantic manifold priors (e.g., shapes and textures) can be leaked from robust models (Engstrom et al., 2019).

- **Reveal new insights of manifold feedback during query-efficient zeroth-order search.** In practice, the data manifold is difficult to characterize. We propose the use of three proxies for manifold distance, which all show consistent results in terms of an adversary's ability to search near the manifold. This methodology allows us to empirically demonstrate the connection between dimension reduction, model robustness, and manifold feedback from the model, beyond the known convergence rates tied to dimensionality (Nesterov & Spokoiny, 2017). Our findings inform how to search closer to the manifold (Table 1), reduce gradient deviation (Table 2), and improve query efficiency (Figure 2) in a simple and generic way for hard-label attacks.

- **Attack-agnostic method for super-pixel grouping.** We show that spatial dimension reduction of a decision-based gradient estimate acts as an attack- and knowledge-agnostic method for searching over super-pixels of an image. More importantly, this helps an attacker exploit a model's reaction to salient input changes, leading to samples closer to the manifold compared to the attack on full dimension. As a result, we demonstrate up to 200% and 340% success rate improvement for state-of-the-art hard-label attacks HSJA (Chen et al., 2019) and Sign-OPT attack (Cheng et al., 2020), respectively.

## 2 RELATED WORK

Since the original discovery of adversarial samples against deep models (Szegedy et al., 2013; Goodfellow et al., 2014), the prevailing question was why such examples existed. The original assumption was that adversarial examples lived in low-probability pockets of the input space, and were never encountered during parameter optimization (Szegedy et al., 2013). This effect was believed to be amplified by the linearity of weight activations in the presence of small perturbations (Goodfellow et al., 2014). These assumptions were later challenged by the boundary tilting assumption, which in summary 1) asserts that the train and test sets of a model only occupy a sub-manifold of the true data, while the decision boundary lies close to samples on and beyond the sub-manifold (Tanay & Griffin, 2016), and 2) supports the "data geometric" view, where high-dimensional geometry of the true data manifold enables a low-probability error set to exist (Gilmer et al., 2018). Likewise the boundary tilting assumption describes adversarial samples as leaving the manifold, which has inspired defenses based on projecting such samples back to the data manifold (Jalal et al., 2019; Samangouei et al., 2018). However, these approaches were later defeated by adaptive attacks (Carlini et al., 2019; Carlini & Wagner, 2017; Tramer et al., 2020).

We investigate the scenario where an adversary uses zeroth-order information (i.e., top-1 label feed-back) to estimate the desired gradient direction (Cheng et al., 2020; Chen et al., 2019). Contemporary attacks in this setting are variants of random gradient-free method (RGF) (Nesterov & Spokoiny, 2017), and rely on formulations which convert the top-1 (hard) label, which is a step function, into a continuous real-valued function $g : \mathbb{R}^d \to \mathbb{R}$, which takes search direction $\boldsymbol{\theta} \in \mathbb{R}^d$ and outputs the distance to the nearest adversarial example (Cheng et al., 2018). The gradient estimate is conceived as a function of the gradient $\nabla g$ and can be estimated with either two samples of information (Sign-OPT) (Cheng et al., 2020), or a single point (HopSkipJumpAttack) (Chen et al., 2019). Details of specific formulations for each attack are provided in Section A.2 of the Appendix.

Query efficiency is a persistent desire in the study of hard-label attacks. One clue for achieving efficiency comes from the theory of gradient estimation error and convergence, which shows that the estimation cost is polynomial in $d$, the dimension of the optimized variable, thus motivating the use of standard dimension-reduction techniques (Tu et al., 2019). However, to date it is not completely understood how this relates to traversal through the data manifold. We leverage previous results of the gradient-level setting (Stutz et al., 2019; Engstrom et al., 2019) to formulate an explanation of manifold leakage during hard-label adversarial attacks.

## 3 NOISY MANIFOLD DISTANCE ORACLE

Santurkar et al. (2019) demonstrate that the gradients of robust models have higher visual semantic alignment with the data compared to gradients of standard models. We build on this finding by first assuming that the benign observable data generates from a true lower-dimension distribution. Under the boundary-tilting assumption, this lower-dimension distribution forms a manifold onto which new observations, either benign or adversarial, can be encoded (Tanay & Griffin, 2016). Likewise, we assume that deep learning models will learn a lower-dimension representation of the observable data, e.g., feature layers of convolutional neural networks learn to encode training observations onto a low dimension *approximate* manifold (Zhang et al., 2018). When an adversary creates adversarial samples, they are leveraging a pathway that shadows the model gradient, not the true manifold. Thus there is the possibility that adversarial samples are considered "off-manifold", e.g., cannot be expected to generate naturally from the true manifold. However, it is critical for adversarial samples to be as close to the manifold as possible, since on-manifold adversarial examples can exploit the fundamental generalization error of the model (Stutz et al., 2019). More formally, we define the notion of *manifold distance* as follows.

**Definition 3.1** (Manifold Distance). Consider the benign sample $\mathbf{x}_0$ and adversarial counterpart $\mathbf{x}$. Assuming a perfect encoding back to the true manifold $\phi$, the manifold distance is defined as $\mathrm{d}(\phi(\mathbf{x}_0), \phi(\mathbf{x}))$, where $\mathrm{d}$ is a distance function with the domain of the true manifold.

Unfortunately, unless the true manifold for a dataset is known, it is impossible to define $\phi$. Instead, a proxy $\mathrm{d}'$ can be used such that $\mathrm{d}'(\mathbf{x}, \mathbf{x}') \sim \mathrm{d}(\phi(\mathbf{x}), \phi(\mathbf{x}'))$. In practice, one can implement $\mathrm{d}'$ with any perceptual distance score, such as Learned Perceptual Image Patch Similarity (Zhang et al., 2018). If relying on a distance measure $\mathrm{d}$, such as the $L_p$-norm, an approximate encoder $\phi'(\cdot) \sim \phi(\cdot)$ can be learned using reconstruction-based training of autoencoders (Stutz et al., 2019), or leveraging feature layers of convolutional neural networks (Zhang et al., 2018). We are interested in the class of hard-label adversaries that implicitly minimize some proxy of the manifold distance. Given the result of Santurkar et al. (2019), the robust model's gradient could be treated as a *manifold distance oracle*, because it leaks the direction towards its approximate manifold. As a result, the model acts as an oracle responding to queries about manifold distance, or in other words, an implicit proxy for manifold distance, $\mathrm{d}'$. In the hard-label setting, the data manifold, true gradient, and model parameters are not accessible. Thus we are interested in a decision-based version of the manifold distance oracle, defined as follows.

**Definition 3.2** (Noisy Manifold Distance Oracle). Consider a manifold distance oracle instantiating $\mathrm{d}'$, benign sample $\mathbf{x}_0$, and pair of adversarial samples $(\mathbf{x}', \mathbf{x}'')$ such that $\mathrm{d}'(\mathbf{x}_0, \mathbf{x}') < \mathrm{d}'(\mathbf{x}_0, \mathbf{x}'')$, e.g., $\mathbf{x}'$ is considered on-manifold while $\mathbf{x}''$ is not. In the hard-label setting, the noisy manifold distance (NMD) oracle instantiates $\mathrm{d}''$ such that $\mathrm{d}''(\mathbf{x}_0, \mathbf{x}') = 0$ and $\mathrm{d}''(\mathbf{x}_0, \mathbf{x}'') = 1$.

During a hard-label attack, the adversary searches in a direction that minimizes perceptual distance to the original sample. Concurrently, the adversary can be said to implicitly minimize the expected

output of the NMD oracle, which is a binary indicator that a sample is on-manifold or not. Without knowledge of the true (or approximate) manifold, this requires careful selection of the search direction from the current sample. Since the search direction of contemporary hard-label attacks is synthesized over expectation of a ball around the adversarial sample, we are interested in search directions such as $\mathbf{x}_0 - \mathbf{x}'$ which minimize the expected distance to the manifold.

To formalize the entailed information in the NMD oracle, we turn to a standard result in data processing, which states the following:

**Definition 3.3** (Data Processing Inequality (DPI) (Beaudry & Renner, 2012)). If three random variables form the Markov chain $X \to Y \to Z$, then their mutual information (MI) has the relation $I(X;Y) \geqslant I(X;Z)$.

We assume the data manifold $\mathcal{M}$, the input gradient $\mathcal{G}$, and the hard-label gradient estimate $\ddot{\mathcal{G}}$ will form the Markov chain $\mathcal{M} \to \mathcal{G} \to \ddot{\mathcal{G}}$. This assumption is reasonable due to the observations by Santurkar et al. (2019); modifying the sampled data manifold (e.g., by adding adversarial samples through saddle-point optimization) causally induces a smoother loss surface, which imposes its own gradient distribution. Likewise, the true gradient and gradient estimate of hard-label attack are causally linked due to the estimate's bounded variance (Cheng et al., 2020).

If $I(\mathcal{M},\mathcal{G})$ is larger for adversarially robust models, by Definition 3.3 the upper bound on $I(\mathcal{M},\ddot{\mathcal{G}})$ is larger, which means more manifold information could be leaked in the noisy gradient. This information could be used to search in the direction where $d''$ is minimized in expectation, leading towards on-manifold examples. However, DPI only offers an upper bound, thus the distance decrease is not guaranteed, only suggested. In the information theoretic sense, does this mean the gradients of models robust in an $\epsilon$-ball around each sample can reveal more information about the distance to training data than standard models? An immediate follow-up concern is whether other factors can influence the model to reveal this information, such as the problem dimensionality. As a first step we posit the following hypothesis:

**Hypothesis 1.** *Consider the manifold distribution $\mathcal{M}$ which can generate data to train a natural model with gradient distribution $\mathcal{G}$, and train robust model with smoothed gradient distribution $\mathcal{G}'$. We posit that their manifold-gradient mutual information $I$ has the relation $I(\mathcal{M},\mathcal{G}') \geq I(\mathcal{M},\mathcal{G})$.*

In order to empirically verify Hypothesis 1, we must parameterize the notion of model robustness while solving for $I(\mathcal{M},\mathcal{G})$, given an arbitrary gradient distribution $\mathcal{G}$ and manifold distribution $\mathcal{M}$. Schmidt et al. (2018) have shown that robust training requires additional data as a function of the data dimensionality. We leverage the data model and results from Schmidt et al. (2018) to derive an analytical solution for $I(\mathcal{M},\mathcal{G})$, since we can parameterize model robustness as a function of data size and dimensionality. Consequently, the remainder of our theoretical analysis assumes a Gaussian mixture data model.

**Definition 3.4** (Data model and optimal weights (Schmidt et al., 2018)). Let $\boldsymbol{\mu} \in \mathbb{R}^d$ be the per-class centers (means) and let $\sigma > 0$ be the variance parameter. Then the $(\boldsymbol{\mu}, \sigma \boldsymbol{I})$-Gaussian model is defined by the following distribution over $(\mathbf{x},y) \in \mathbb{R}^d \times \{\pm 1\}$: First, draw a label $y \in \{\pm 1\}$ uniformly at random. Then sample the data point $\mathbf{x} \in \mathbb{R}^d$ from $\mathcal{N}(y \cdot \boldsymbol{\mu}, \sigma \boldsymbol{I})$.

**Definition 3.5** (Optimal classification weight (Schmidt et al., 2018)). Fix $\sigma \leq c_1 d^{\frac{1}{4}}$ for the universal constant $c_1$, and samples $(\mathbf{x}_1, y_1), \cdots, (\mathbf{x}_n, y_n)$ drawn $i.i.d$ from the $(\boldsymbol{\mu}, \sigma \boldsymbol{I})$-Gaussian model with $||\boldsymbol{\mu}|| = \sqrt{d}$ (i.e., $\boldsymbol{\mu}_k = 1$ for all dimensions $k \in \{0, \ldots, d\}$). Schmidt et al. (2018) prove that the weight setting $\widehat{\mathbf{w}} = \frac{1}{n} \sum_{i=1}^{n} y_i \mathbf{x}_i$ yields an $l_\infty^\epsilon$-robust classification error of at most 1% for the linear classifier $f_{\widehat{\mathbf{w}}} : \mathbb{R}^d \to \{\pm 1\}$ instantiated as $f_{\widehat{\mathbf{w}}}(x) = \text{sign}(\widehat{\mathbf{w}}^T \mathbf{x})$ if

$$n \geq \begin{cases} 1, & \text{for } \epsilon \leq \frac{1}{4} d^{-\frac{1}{4}} \\ c_2 \epsilon^2 \sqrt{d}, & \text{for } \frac{1}{4} d^{-\frac{1}{4}} \leq \epsilon \leq \frac{1}{4} \end{cases}, \tag{1}$$

for a universal constant $c_2$.

Note that the instantiation of $\widehat{\mathbf{w}}$ must change with choice of $\epsilon$ and $d$. We can leverage the weight settings as a function of $n$ and $d$ to give a definition of manifold-gradient mutual information.

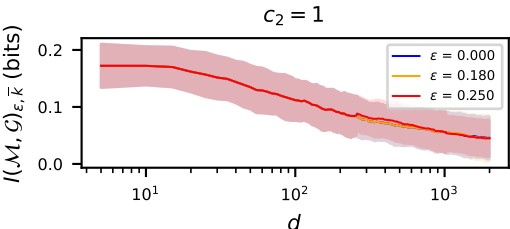 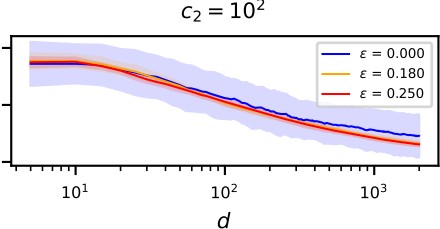

Figure 1: a) Average per-dimension mutual information ($I$) over dimension $d$ for values of $c_2$ and $\epsilon$ in Equation 13, log-scale $d$-axis with $d \in [5, 2000)$, average over ten seeds. The approximate mutual information is higher for robust and standard models at lower $d$ regardless of $c_2$ and choice of $\epsilon$.

### 3.1 MANIFOLD-GRADIENT MUTUAL INFORMATION

Notice the classifier $\text{sgn}(\cdot)$ in Definition 3.5 is discontinuous at $\mathbf{x}_k = 0$ for any dimension $k$. Instead we consider the sub-gradient of the classifier at $\mathbf{x}_k < 0$ and $\mathbf{x}_k > 0$. In either case (non-robust or robust), the input sub-gradient for $f_{\widehat{w}}(\mathbf{x}'_k)$ is defined dimension-wise for our isotropic Gaussian as $\nabla_{\mathbf{x}'_k} f_{\widehat{\mathbf{w}}_k} = \text{sign} \, \mathbf{w}_k$. Since the weight of each dimension is Gaussian distributed with $\widehat{\mathbf{w}}_k \sim \mathcal{N}(\boldsymbol{\mu}_k, \sigma^2)$, we can define the distribution of gradients as $\mathcal{G} \sim \text{Rademacher}\,(\mathbb{P}_{\widehat{\mathbf{w}}_k \sim \mathcal{N}} [\widehat{\mathbf{w}}_k \geq 0])$. Using this fact, we define manifold-gradient mutual information in three parts: 1) defining the manifold-gradient point-wise joint probabilities between $\mathbf{g}_k$ and $\mathbf{x}_k$ at each dimension $k$ for the sub-gradient cases where $\mathbf{x}_k > 0$ and $\mathbf{x}_k < 0$, 2) defining the manifold-gradient marginal probability under the gradient, and 3) the marginal probability under the manifold. The complete derivation of the joint and marginal probabilities can be found in Section A.1 of the Appendix. The three parts are used in the standard definition of mutual information (Cover & Thomas, 2006).

**Notation.** Fix $\sigma = c_1 d^{\frac{1}{4}}$ for both cases. We denote the sub-manifold sampled from the positive ($y = 1$) and negative ($y = -1$) classes as $\mathcal{M}^+$ and $\mathcal{M}^-$, respectively. For brevity we label $\mathbf{x}_k > 0$ as $\mathbf{x}^+$ and $\mathbf{x}_k < 0$ as $\mathbf{x}^-$.

**Definition 3.6** (Manifold-Gradient Mutual Information). We define the manifold-gradient mutual information, based on the standard definition of mutual information from information theory (Cover & Thomas, 2006), as

$$
I(\mathcal{M}, \mathcal{G})_{\epsilon, k} = 2 \int_{\mathcal{M}^+} p(1, \mathbf{x}^+) \log(\frac{p(1, \mathbf{x}^+)}{p_{\mathcal{G}}(1) p_{\mathcal{M}}(\mathbf{x}^+)}) \, d\mathbf{x}^+ + 2 \int_{\mathcal{M}^+} p(-1, x^+) \log(\frac{p(-1, x^+)}{p_{\mathcal{G}}(-1) p_{\mathcal{M}}(x^+)}) \, dx^+.
\tag{2}
$$

with the total unnormalized mutual information defined as the summation over dimensions (due to dimension co-independence) $I(\mathcal{M}, \mathcal{G})_\epsilon = \sum_{k=1}^d I(\mathcal{M}, \mathcal{G})_{\epsilon, k}$.

### 3.2 MUTUAL INFORMATION AS A FUNCTION OF DIMENSIONALITY

To provide numerical support for Hypothesis 1, we run experiments using the Riemann approximation of Equation 13, provided in the Appendix as Equation 15. We estimate the average per-dimension mutual information, $I(\mathcal{M}, \mathcal{G})_{\epsilon, \bar{k}} = \frac{I(\mathcal{M}, \mathcal{G})_\epsilon}{d}$, for the case where $\mathbf{x} \in \mathbb{R}^d$ while varying the dimensionality term $d \in [5, 2000)$ against values of $c_2 \in \{1, 100\}$ and $\epsilon \in \{0.000, 0.180, 0.250\}$. The values of $c_2$ represent two multiplicative factors for number of samples in robust models (Equation 1). In our experiments, we target an error within $10^{-1}$ (e.g., $0.9 \leq p_{\mathcal{G}}(1) + p_{\mathcal{G}}(-1) \leq 1.0$). Thus we multiply each branch of Equation 1 by a large constant ($10^4$). We run the approximation over ten different random seeds and show the average with standard error shaded.

The estimation result is shown in Figure 1 with log-scale x-axis. Regardless of $c_2$ and $\epsilon$, lower values of the dimensionality evidence a higher mutual information. We minimize variance of the estimate when $c_2 = 10^2$ (right plot shaded area), which follows intuition due to the higher sample count in the estimate.

**Observation 1.** *Given reduced data dimensionality, a robust model could increase $I(\mathcal{M}, \mathcal{G})_{\epsilon, \overline{k}}$ and lead to leaking better search direction through the gradient (e.g., act as manifold distance oracle). This supports Hypothesis 1.*

This can theoretically explain the high visual alignment observed empirically by Engstrom et al. (2019) and Santurkar et al. (2019) on robust models. From the security perspective, the NMD oracle acts as a side channel leaking sensitive information as a factor of the model robustness and data dimensionality.

## 4    Zeroth-order Search Through the Manifold Distance Oracle

According to Observation 1, the true gradient and manifold of a robust model have higher mutual information, and this is exacerbated by reducing the data dimensionality. Under our Markov chain assumption, this means an attack algorithm can act as a noisy manifold distance oracle, and this oracle could be upper bounded by the true gradient-manifold mutual information. Although the data dimensionality and robustness are controlled by the model designer, an attacker can search in arbitrarily lower dimensionality through dimension-reduction techniques, such as autoencoder-based attacks (Tu et al., 2019). In fact, in the image domain the intrinsic dimensionality of data can be lower than the true dimension (Amsaleg et al., 2017). In order to connect the notion of manifold-gradient mutual information with on-manifold adversarial samples of real datasets, we posit the following.

**Hypothesis 2.** *Consider Observation 1 and Definition 3.3 (DPI), then due to the higher upper bound on $I(\mathcal{M}, \ddot{\mathcal{G}})$ and leaking better search directions, a hard-label adversary can minimize $d''$ in expectation on robust models when the gradient estimate dimensionality is reduced.*

In the most common problem setting, the adversary is interested in attacking a $K$-way multi-class classification model $f : \mathbb{R}^d \to \{1, \dots, K\}$. Given an original example $\mathbf{x}_0$, the goal is to generate adversarial example $\mathbf{x}$ such that $\mathbf{x}$ is close to $\mathbf{x}_0$ and $f(\mathbf{x}) \neq f(\mathbf{x}_0)$, where closeness is often approximated by the $L_p$-norm of $\mathbf{x} - \mathbf{x}_0$. In the gradient-level setting, we require the gradient $\nabla f(\cdot)$. However, in the hard-label setting we are forced to estimate $\frac{\partial f(\mathbf{x})}{\partial \mathbf{x}}$ without access to $\nabla f(\cdot)$, only decision evaluations of $f$. Rather than optimizing the step function $f$, hard-label attacks minimize the continuous function $g(\boldsymbol{\theta})$, which is an estimate of the distance to the nearest decision boundary in the direction $\boldsymbol{\theta}$. We evaluate the effect of dimension reduction on Sign-OPT attack (Cheng et al., 2020) and HopSkipJumpAttack (HSJA) (Chen et al., 2019), as both are considered state-of-the-art in the literature, and rely on minimization of $g(\cdot)$. We provide a brief overview of their formulation in Section A.2 of the Appendix, and leave details to the respective authors' work. Alternative hard-label attacks, such as RayS by Chen & Gu (2020), do not rely on the explicit zeroth-order gradient estimate from the model. This style of attack behaves differently since it can adapt to the problem dimension independent of the true gradient, which we demonstrate in Section A.5.6 of the Appendix.

### 4.1    Dimension-reduced zeroth-order search

To test Hypothesis 2, we modify existing hard-label attacks to produce dimension-reduced variants. This scheme enables dynamic scaling of the effective dimensionality regardless of specific attack formulation. In practice we implement the reduction through an encoding map $\mathcal{E} : \mathbb{R}^d \to \mathbb{R}^{d'}$ for reduced dimension $d'$ and decoding map $\mathcal{D} : \mathbb{R}^{d'} \to \mathbb{R}^d$. In general the adversarial sample is created by $\mathbf{x} = \mathbf{x}_0 + g(\mathcal{D}(\boldsymbol{\theta}')) \frac{\mathcal{D}(\boldsymbol{\theta}')}{||\mathcal{D}(\boldsymbol{\theta}')||}$, where $\boldsymbol{\theta}' \in \mathbb{R}^{d'}$ and is optimized depending on the respective attack (e.g., Sign-OPT and HSJA), and as before, $g$ is a measure of distance to the decision boundary in direction $\mathcal{D}(\boldsymbol{\theta}')$. The mapping functions can be initialized with either an autoencoder (AE), or a pair of channel-wise bilinear transform functions (henceforth referred to as BiLN) which simply scales the spatial dimension of the input up or down. This represents two distinct methods to search over super-pixels of the image, which either rely on an approximate description of the manifold (AE), or instead exploit the known spatial co-dependence of images (BiLN). The implementation and training details of the AE variant can be found in Section A.4.2 of the Appendix. To study the effect of dimension-reduction *without* semantic information, we implement a random variant of BiLN (Rand) which samples a subset of coordinates uniform-randomly from the source image as the dimension-reduced version, then replaces the pixels at these coordinates with those from the gradient

estimate. This is meant to show the effect of discarding some semantic information (e.g., spatial correlation) in the update.

## 4.2 ESTIMATING MANIFOLD DISTANCE

We leverage three proxies of manifold distance in order to test Hypothesis 2. The Learned Perceptual Image Patch Similarity (LPIPS) acts as a proxy for manifold distance, $d'$, and computes a distance that correlated well with human perception in human studies (Zhang et al., 2018; Laidlaw et al., 2021). We use the same LPIPS code and checkpoint provided by the authors. Fréchet Inception Distance (FID) (Heusel et al., 2018) is similar to LPIPS, and leverages the internal representations of deep networks as an approximate encoding onto the manifold. Although FID lacks human studies, Heusel et al. (2018) show it is viable for scoring the visual quality of synthetically generated images, which offers us a comparison against LPIPS. In addition to LPIPS and FID, we create an approximate encoding $\phi'$ by taking the encoder of trained autoencoders for each dataset, which can be used to compute $L_\infty$ distance between encoded samples. In other words, this lets us compute $||\phi'(\mathbf{x}_0) - \phi'(\mathbf{x})||_\infty$ for benign sample $\mathbf{x}_0$ and adversarial sample $\mathbf{x}$. The results on FID and our trained autoencoder were consistent with LPIPS, so they are described in Section A.5.9 of the Appendix.

Finally, if hard-label gradient estimates on real-world data resulted in a sample close to the approximate manifold, we could say the gradient estimates leveraged noisy mutual information, which may be upper bounded by the clean mutual information (Hypothesis 1). This would manifest in a lower gradient deviation, or in other words, the distance between the true gradient and gradient estimate at the first attack step. We can further infer that the adversarial training effectively smooths the *sampled* data manifold (which generates from true manifold) by augmenting perturbed data samples during training. The smoothing yields a well-defined boundary that aligns with salient input changes (Santurkar et al., 2019), and should further lower variance of the gradient estimate compared to natural models, which improves the baseline performance of an attack. We test this by calculating per-pixel gradient deviation $\frac{||\mathbf{g} - \hat{\mathbf{g}}||_2}{H \times W}$ for true gradient $\mathbf{g}$ (in the direction of the adversarial label), first gradient estimate $\hat{\mathbf{g}}$, estimate height $H$, and estimate width $W$. When taking the true input gradient in the direction of the adversarial label, we use the victim model's original criterion to calculate the gradient, which was cross-entropy for all models in our evaluation.

## 5 RESULTS & DISCUSSION

We test Hypothesis 2 by comparing two SotA hard-label attacks with their compatible dimension-reduced variants, against both natural and robust models. First we show empirical evidence of the relationship between manifold distance and dimension-reduced attacks in Section 5.1. Next in Section 5.2, we investigate the result of Section 5.1 from the perspective of reducing error in the gradient estimate. Finally in Section 5.3, we show how these observations inform better attack design.

**Setup.** We perform experiments using CIFAR-10 (Krizhevsky, 2009) and ImageNet (Krizhevsky et al., 2012) for RGB image data. The natural CIFAR-10 network is the same implementation open-sourced by Cheng et al. (2020). The architecture for ImageNet is the Resnet50 network taken from the PyTorch Torchvision library, and the accompanying pre-trained weights act as the natural model.[1] In addition, we leverage the representative adversarial training technique proposed by Madry et al. (2017) (and their $\epsilon = \frac{8}{255} = 0.031$ checkpoints for $L_\infty$ setting) as the robust models for CIFAR-10 and ImageNet. The BiLN variants downscale to $16 \times 16$ for CIFAR-10, and $32 \times 32$ for ImageNet. We use $L_\infty$-norm versions of attacks for all experiments, and the same $\epsilon$ values for natural models as the $L_\infty$ CIFAR-10 and robust ImageNet (hereafter referred to as Madry CIFAR-10 and Madry ImageNet). All attacks run for 25k queries without early stopping on correctly classified samples. For brevity, we only show results for the untargeted case. Additional implementation details, such as hyperparameters and hardware used, can be found in the Appendices (Section A.4). Code for experiments is provided in the supplementary materials.

---

[1] https://pytorch.org/docs/stable/torchvision/models.html

| Attack Variant | Natural CIFAR-10 | Madry CIFAR-10 | Natural ImageNet | Madry ImageNet |
|---|---|---|---|---|
| HSJA | $0.132 \pm 0.098^{\star}$ | $1.335 \pm 0.611$ | $0.257 \pm 0.378$ | $1.249 \pm 0.652$ |
| HSJA+BiLN | $0.252 \pm 0.165\uparrow$ | $1.147 \pm 0.535\downarrow^{\star}$ | $0.170 \pm 0.143\downarrow^{\star}$ | $1.205 \pm 0.711\downarrow^{\star}$ |
| HSJA+Rand | $1.433 \pm 0.747\uparrow$ | $2.384 \pm 0.503\uparrow$ | $1.276 \pm 0.649\uparrow$ | $1.183 \pm 0.596\downarrow$ |
| Sign-OPT | $0.105 \pm 0.081$ | $0.768 \pm 0.408$ | $0.768 \pm 0.872$ | $1.229 \pm 0.771$ |
| Sign-OPT+BiLN | $0.225 \pm 0.146\uparrow$ | $0.849 \pm 0.397\uparrow$ | $0.176 \pm 0.204\downarrow$ | $0.708 \pm 0.461\downarrow$ |
| Sign-OPT+Rand | $0.440 \pm 0.464\uparrow$ | $1.021 \pm 0.593\uparrow$ | $0.356 \pm 0.385\downarrow$ | $0.367 \pm 0.361\downarrow$ |
| Sign-OPT+AE | $0.331 \pm 0.389\uparrow$ | $0.660 \pm 0.302\downarrow$ | $1.034 \pm 0.571\uparrow$ | $1.658 \pm 0.638\uparrow$ |

Table 1: Average LPIPS scores for each attack's set of 200 adversarial samples on CIFAR-10 and ImageNet (lower is better). Arrows denote higher or lower score compared to baseline variant, and starred items indicate highest success rate.

| Med. Benign Local ID | 0.469 | 0.224 | 1.039 | 2.013 |
|---|---|---|---|---|
| Attack Variant | Natural CIFAR-10 | Madry CIFAR-10 | Natural ImageNet | Madry ImageNet |
| HSJA | $6.65 \pm 0.61^{\star}$ | $5.46 \pm 0.06$ | $77.35 \pm 0.04$ | $77.32 \pm 0.00$ |
| HSJA+BiLN | $5.37 \pm 0.69\downarrow$ | $3.86 \pm 0.10\downarrow^{\star}$ | $55.12 \pm 1.37\downarrow^{\star}$ | $56.14 \pm 0.12\downarrow^{\star}$ |
| HSJA+Rand | $11.33 \pm 7.41\uparrow$ | $2.01 \pm 1.65\downarrow$ | $72.19 \pm 59.98\downarrow$ | $3.22 \pm 2.73\downarrow$ |
| Sign-OPT | $3.72 \pm 0.99$ | $0.71 \pm 0.38$ | $1.70 \pm 1.01$ | $0.55 \pm 0.18$ |
| Sign-OPT+BiLN | $3.71 \pm 1.02\downarrow$ | $0.78 \pm 0.35\uparrow$ | $1.83 \pm 0.97\uparrow$ | $1.74 \pm 0.56\uparrow$ |
| Sign-OPT+Rand | $8.21 \pm 6.67\uparrow$ | $2.32 \pm 2.07\uparrow$ | $37.54 \pm 46.20\uparrow$ | $6.72 \pm 1.54\uparrow$ |
| Sign-OPT+AE | $4.66 \pm 0.86\uparrow$ | $2.48 \pm 0.32\uparrow$ | $36.83 \pm 0.15\uparrow$ | $36.87 \pm 0.31\uparrow$ |

Table 2: Average per-pixel gradient deviation on natural and robust CIFAR-10 (unit of $10^{-2}$) and ImageNet (unit of $10^{-4}$) over 200 samples. Top row lists the median Local Intrinsic Dimensionality (LID) of benign samples from the dataset. Arrows denote higher or lower deviation compared to baseline variant, and starred items indicate highest success rate.

## 5.1 MANIFOLD DISTANCE

LPIPS results are shown in Table 1, with colored arrows denoting either lower distance than baseline variant (green arrow), or a higher distance (red arrow). Generally, the dimension-reduced variants lower the proxy of manifold distance on ImageNet more often than on CIFAR-10 (green arrows). The random sampling variant ($\star$-Rand) discards the semantic priors of the estimate, and in fact it achieved the lowest SR AUC scores, despite having lower scores. Our results using LPIPS are consistent with $L_{\infty}$ distance of the manifold approximation (Section A.5.9), and Fréchet Inception Distance (Section A.5.8), which all demonstrate a tendency to be lower with dimension-reduced attacks.

**Observation 2.** *Dimension-reduced hard-label attacks can have lower LPIPS score, $L_{\infty}$ approximated distance, and Fréchet Inception Distance (and thus lower manifold distance) on robust models if they preserve semantic priors in the update, which supports Hypothesis 2.*

## 5.2 GRADIENT DEVIATION

The results for gradient deviation are shown in Table 2. Notably, an attack can have high gradient deviation despite low LPIPS score (AE case, bottom row). Likewise, low deviation does not imply successful attack, as we show later with the Rand variant (rows three and six). We investigated why Madry ImageNet did not always have lower gradient deviation, which we posit is due to having a higher true dimensionality. For the benign samples of each dataset we estimated the Local Intrinsic Dimensionality (LID), which was proposed to estimate true data dimensionality in a region around samples (Amsaleg et al., 2017). In the top row of Table 2 we find the median LID is similar between natural and robust CIFAR-10, but much higher on robust ImageNet than natural. Since our results of Section 3 suggested that higher problem dimension reduced mutual information, we suspect the Madry ImageNet model *reduces* the leakage through the NMD oracle through higher true data dimensionality. We leave a deeper analysis of this direction for future work. Results on additional

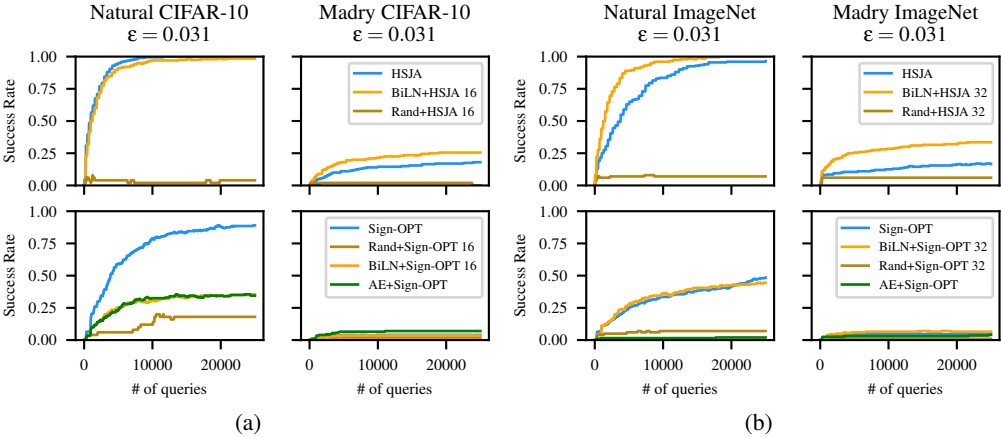

Figure 2: Success rates across attacks over 200 samples on CIFAR-10 (a) and ImageNet (b).

robust CIFAR-10 models are provided in Section A.5.2 of the Appendix, which exhibited a similar trend of lower gradient deviation. Sign-OPT has a universally lower gradient deviation than HSJA, which aligns with findings of Liu et al. (2020).

**Observation 3.** *The gradient deviation is universally lower on the robust CIFAR-10 model for BiLN attacks (rows two and five). For ImageNet, deviation on robust models is either lower or similar (rows one, two, four, and seven).*

### 5.3 INFORMING PRACTICE

We have shown that dimension reduction has unexpected consequences in terms of manifold distance, and on CIFAR-10 and some ImageNet cases, leads to a lower gradient deviation on the robust model. We finalize our contribution by providing a comprehensive evaluation of the attack success rates in Figure 2 against number of queries. The plots are quantified by taking their max-normalized Trapezoid rule area-under-curve (AUC).[2] For comparison, the highest AUC scores are starred in the previous tables. Our dimension-reduced HSJA+BiLN variant (yellow line) surpasses the previous SotA hard-label attack for ImageNet, HSJA, on both natural and robust models. This variant also exhibited the lowest LPIPS score across attack variants. However, lowest LPIPS score does not imply highest SR, evidenced with HSJA+Rand on natural ImageNet (brown line, AUC= 0.077) and Sign-OPT variants on either dataset (e.g., yellow line in Madry ImageNet, AUC= 0.215). Low gradient deviation does not imply higher attack success, evidenced by Sign-OPT+BiLN in Table 2 for Madry CIFAR-10 (AUC= 0.156) or HSJA+Rand and Sign-OPT+Rand (AUC= 0.088 and AUC= 0.092, respectively). The Rand variants, combined with our findings so far, allow us to say the following.

**Observation 4.** *Successful attacks exhibit preservation of leaked semantic priors. Measures of manifold distance such as LPIPS tend to be lower on dimension-reduced attacks, independent of variance in the gradient estimate.*

We posit that minimizing gradient deviation through correction of estimator bias alone could be misleading, since the semantic information provided by a better NMD oracle (due to dimension reduction) can potentially improve the gradient deviation. Although our theoretical analysis focuses on robust models, we suspect future hard-label attacks may treat $\epsilon$ as a useful prior, which carries with it implications about when to deploy robust models in society. On the contrary, natural models will respond to any input changes, even if they are semantically meaningless (Santurkar et al., 2019), so depending on the adversary's goal (e.g., evasion or information leakage), they could be less useful in the hard-label setting.

---

[2]Tables with all normalized SR AUC scores are provided in Section A.5.3 of the Appendix.

## 6 CONCLUSION

Despite the recent progress in zeroth-order attack methods, open questions remain about their precise behavior. We develop an information-theoretic analysis that sheds light on their ability to produce on-manifold adversarial examples. Through experiments on real-world datasets, we show an over two-fold increase in attack success rates by leveraging new findings about manifold distance and gradient deviation. With knowledge of the manifold-gradient relationship, it is possible to further refine hard-label attacks, and inform a better evaluation of model robustness. Given the availability of larger datasets in the future, our method may turn the strength of deep learning, which is efficiently extracting patterns in large-scale data, into a weakness.

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

## A    APPENDIX

### A.1    DERIVATION OF MANIFOLD-GRADIENT MUTUAL INFORMATION (MI)

We define the manifold-gradient point-wise joint probability in a case-wise manner, for the respective values under $\mathbf{g} \in \{-1, 1\}^d$ and $\mathbf{x} \in \mathbb{R}^d$. We are concerned with the sub-gradient cases where $\mathbf{x} > 0$ (denoted $\mathbf{x}^+$) and $\mathbf{x} < 0$ (denoted $\mathbf{x}^-$) which correspond to fixed values of $\mathbf{g}$ based on class means $y \cdot \boldsymbol{\mu}$ with $y \in \{-1, 1\}$. This gives for each dimension $k$,

$$p(\mathbf{g}_k = 1, \mathbf{x}_k^+) = \frac{1}{2\sigma\sqrt{2\pi}}\exp\left(-\frac{1}{2}\left(\frac{\mathbf{x}_k^+ - \boldsymbol{\mu}_k}{\sigma}\right)^2\right)$$
$$p(\mathbf{g}_k = 1, \mathbf{x}_k^-) = \frac{1}{2\sigma\sqrt{2\pi}}\exp\left(-\frac{1}{2}\left(\frac{\mathbf{x}_k^+ + \boldsymbol{\mu}_k}{\sigma}\right)^2\right) \tag{3}$$

$$p(\mathbf{g}_k = -1, \mathbf{x}_k^+) = \frac{1}{2\sigma\sqrt{2\pi}}\exp\left(-\frac{1}{2}\left(\frac{\mathbf{x}_k^+ + \boldsymbol{\mu}_k}{\sigma}\right)^2\right)$$
$$p(\mathbf{g}_k = -1, \mathbf{x}_k^-) = \frac{1}{2\sigma\sqrt{2\pi}}\exp\left(-\frac{1}{2}\left(\frac{\mathbf{x}_k^+ - \boldsymbol{\mu}_k}{\sigma}\right)^2\right). \tag{4}$$

Since the Schmidt et al. Gaussian mixture is created symmetrically (the probability mass is evenly split between the two classes i.e., the mixture comprises one Gaussian offset by $\boldsymbol{\mu}_k$ and mirrored at $\mathbf{x}_k = 0$) we can simplify to

$$p(\mathbf{g}_k = 1, \mathbf{x}_k^+) = \frac{1}{2\sigma\sqrt{2\pi}}\exp\left(-\frac{1}{2}\left(\frac{\mathbf{x}_k^+ - \boldsymbol{\mu}_k}{\sigma}\right)^2\right), \tag{5}$$

$$p(\mathbf{g}_k = -1, \mathbf{x}_k^+) = \frac{1}{2\sigma\sqrt{2\pi}}\exp\left(-\frac{1}{2}\left(\frac{\mathbf{x}_k^+ + \boldsymbol{\mu}_k}{\sigma}\right)^2\right), \tag{6}$$

where $\mathbf{x} \sim \mathcal{N}(y \cdot \boldsymbol{\mu}, \sigma \boldsymbol{I})$. In words, Equation 6 is the symmetrical tail of the Gaussian mixture marginal while Equation 5 is the remainder of the mixture.

Similarly, a point-wise gradient is given as the Rademacher outcome $\mathbf{g}_k \in \{\pm 1\}$. The choice of $\epsilon$ directly influences the marginal probability over the manifold. The marginal probability over the manifold can be given as the Riemann approximations

$$p_{\mathcal{G}}(\mathbf{g}_k = 1)_\epsilon = \frac{1}{2\sigma\sqrt{2\pi}}\sum_{i=1}^n \exp\left(-\frac{1}{2}\left(\frac{\mathbf{x}_{i,k}^* - \boldsymbol{\mu}_k}{\sigma}\right)^2\right)\Delta_i \tag{7}$$

and

$$p_{\mathcal{G}}(\mathbf{g}_k = -1)_\epsilon = \frac{1}{2\sigma\sqrt{2\pi}}\sum_{i=1}^n \exp\left(-\frac{1}{2}\left(\frac{\mathbf{x}_{i,k}^* + \boldsymbol{\mu}_k}{\sigma}\right)^2\right)\Delta_i, \tag{8}$$

with $\Delta_i = \mathbf{x}_{i,k}^+ - \mathbf{x}_{i-1,k}^+$ for arbitrary $\mathbf{x}_{i,k}^* \in [\mathbf{x}_{i-1,k}^+, \mathbf{x}_{i,k}^+]$, and $n$ is controlled by the hyper-parameter $\epsilon$.

The marginal for the manifold under the gradient is given similarly as

$$p_{\mathcal{M}}(\mathbf{x}_k) = p(\mathbf{g}_k = 1, \mathbf{x}_k^+) + p(\mathbf{g}_k = -1, \mathbf{x}_k^+)$$
$$= \frac{1}{\sigma\sqrt{2\pi}}\exp\left(-\frac{1}{2}\left(\frac{\mathbf{x}_k^+ - \boldsymbol{\mu}_k}{\sigma}\right)^2\right) + \frac{1}{\sigma\sqrt{2\pi}}\exp\left(-\frac{1}{2}\left(\frac{\mathbf{x}_k^+ + \boldsymbol{\mu}_k}{\sigma}\right)^2\right), \tag{9}$$

where $\mathbf{x}_k^+ > 0$ for all dimensions $k$. Denote the sub-manifold sampled from the positive ($y = 1$) and negative ($y = -1$) classes as $\mathcal{M}^+$ and $\mathcal{M}^-$, respectively. Our definition for manifold-gradient mutual information is based on the standard definition of mutual information from information theory (Cover & Thomas, 2006),

$$I(\mathcal{M}, \mathcal{G})_{\epsilon,k} = \int_{\mathcal{M}} \int_{\mathcal{G}} p(\mathbf{g}_k, \mathbf{x}_k) \log(\frac{p(\mathbf{g}_k, \mathbf{x}_k)}{p_{\mathcal{G}}(\mathbf{g}_k)p_{\mathcal{M}}(\mathbf{x}_k)}) \, d\mathbf{g}_k \, d\mathbf{x}_k, \tag{10}$$

where $\epsilon$ is treated as a hyper-parameter controlling the value of $n$ in $p_{\mathcal{G}}(\mathbf{g}_k)$. By substitution into Equation 10 we have

$$\begin{aligned} I(\mathcal{M}, \mathcal{G})_{\epsilon,k} = & \int_{\mathcal{M}} p(1, \mathbf{x}_k) \log(\frac{p(1, \mathbf{x}_k)}{p_{\mathcal{G}}(1)p_{\mathcal{M}}(\mathbf{x}_k)}) \, d\mathbf{x}_k \\ & + \int_{\mathcal{M}} p(-1, \mathbf{x}_k) \log(\frac{p(-1, \mathbf{x}_k)}{p_{\mathcal{G}}(-1)p_{\mathcal{M}}(\mathbf{x}_k)}) \, d\mathbf{x}_k. \end{aligned} \tag{11}$$

This is split further similar to true positive, true negative, false positive, and false negative, as

$$\begin{aligned} I(\mathcal{M}, \mathcal{G})_{\epsilon,k} = & \int_{\mathcal{M}^+} p(1, \mathbf{x}_k^+) \log(\frac{p(1, \mathbf{x}_k^+)}{p_{\mathcal{G}}(1)p_{\mathcal{M}}(\mathbf{x}_k^+)}) \, d\mathbf{x}_k^+ \\ & + \int_{\mathcal{M}^-} p(1, \mathbf{x}_k^-) \log(\frac{p(1, \mathbf{x}_k^-)}{p_{\mathcal{G}}(1)p_{\mathcal{M}}(\mathbf{x}_k^-)}) \, d\mathbf{x}_k^- \\ & + \int_{\mathcal{M}^+} p(-1, \mathbf{x}_k^+) \log(\frac{p(-1, \mathbf{x}_k^+)}{p_{\mathcal{G}}(-1)p_{\mathcal{M}}(\mathbf{x}_k^+)}) \, d\mathbf{x}_k^+ \\ & + \int_{\mathcal{M}^-} p(-1, \mathbf{x}_k^-) \log(\frac{p(-1, \mathbf{x}_k^-)}{p_{\mathcal{G}}(-1)p_{\mathcal{M}}(\mathbf{x}_k^-)}) \, d\mathbf{x}_k^-, \end{aligned} \tag{12}$$

and simplified due to symmetry at 0 as

$$\begin{aligned} I(\mathcal{M}, \mathcal{G})_{\epsilon,k} = & 2 \int_{\mathcal{M}^+} p(1, \mathbf{x}_k^+) \log(\frac{p(1, \mathbf{x}_k^+)}{p_{\mathcal{G}}(1)p_{\mathcal{M}}(\mathbf{x}_k^+)}) \, d\mathbf{x}_k^+ \\ & + 2 \int_{\mathcal{M}^+} p(-1, \mathbf{x}_k^+) \log(\frac{p(-1, \mathbf{x}_k^+)}{p_{\mathcal{G}}(-1)p_{\mathcal{M}}(\mathbf{x}_k^+)}) \, d\mathbf{x}_k^+. \end{aligned} \tag{13}$$

The total un-normalized mutual information is given by the summation over dimensions $I(\mathcal{M}, \mathcal{G})_\epsilon = \sum_{k=1}^{d} I(\mathcal{M}, \mathcal{G})_{\epsilon,k}$. Notably the cases for each possible scenario under detection theory are represented. Each case is bounded by the results of Schmidt et al. (2018). By substitution from each marginal and joint probability in Equations 8, 3, and 4 respectively, we have the closed form solution for mutual information.

This leads to the Riemann approximation of Equation 13,

$$\begin{aligned} I(\mathcal{M}, \mathcal{G})_{\epsilon,k} = & 2 \sum_{i=1}^{n} p(1, \mathbf{x}_{i,k}^*) \log(\frac{p(1, \mathbf{x}_{i,k}^*)}{p_{\mathcal{G}}(1)p_{\mathcal{M}}(\mathbf{x}_{i,k}^*)}) \Delta_i \\ & + 2 \sum_{i=1}^{n} p(-1, \mathbf{x}_{i,k}^*) \log(\frac{p(-1, \mathbf{x}_{i,k}^*)}{p_{\mathcal{G}}(-1)p_{\mathcal{M}}(\mathbf{x}_{i,k}^*)}) \Delta_i. \end{aligned} \tag{14}$$

with $\Delta_i = \mathbf{x}_{i,k}^+ - \mathbf{x}_{i-1,k}^+$ for arbitrary positive $\mathbf{x}_{i,k}^* \in [\mathbf{x}_{i-1,k}^+, \mathbf{x}_{i,k}^+]$. Since $\mathbf{x}^+$ is a standard multivariate Gaussian (Cover & Thomas, 2006), the final mutual information is the summation over each dimension,

$$I(\mathcal{M}, \mathcal{G})_\epsilon = 2 \sum_{k=1}^{d} \sum_{i=1}^{n} p(1, \mathbf{x}_{i,k}^*) \log\left(\frac{p(1, \mathbf{x}_{i,k}^*)}{p_\mathcal{G}(1) p_\mathcal{M}(\mathbf{x}_{i,k}^*)}\right) \Delta_i$$
$$+ 2 \sum_{k=1}^{d} \sum_{i=1}^{n} p(-1, \mathbf{x}_{i,k}^*) \log\left(\frac{p(-1, \mathbf{x}_{i,k}^*)}{p_\mathcal{G}(-1) p_\mathcal{M}(\mathbf{x}_{i,k}^*)}\right) \Delta_i. \tag{15}$$

## A.2 HARD-LABEL ATTACK FORMULATION

Contemporary hard-label attacks are variants of random gradient-free method (RGF) (Nesterov & Spokoiny, 2017), a gradient estimator which yields the estimate $\hat{\mathbf{g}}$ over $q$ random directions $\{\mathbf{u}_i\}_{i=1}^{q}$.

**OPT-Attack** For benign example $\mathbf{x}_0$, true label $y_0$, and hard-label black-box function $f : \mathbb{R}^d \to \{1, \dots, K\}$, Cheng et al. (2019) define the objective function $g : \mathbb{R}^d \to \mathbb{R}$ as a function of search direction $\boldsymbol{\theta}$, where the optimal solution is $g(\boldsymbol{\theta}^*)$, the minimum distance from $\mathbf{x}_0$ to the nearest adversarial example along the direction $\boldsymbol{\theta}^*$. For the untargeted attack, $g(\boldsymbol{\theta})$ is the distance to any decision boundary along direction $\boldsymbol{\theta}$, and allows for estimating the gradient as

$$\hat{\mathbf{g}} = \frac{1}{q} \sum_{i=0}^{q} \frac{g(\boldsymbol{\theta} + \beta \mathbf{u}_i) - g(\boldsymbol{\theta})}{\beta} \cdot \mathbf{u}_i, \tag{16}$$

where $\beta$ is a small smoothing parameter. Notably, $g(\boldsymbol{\theta})$ is continuous even if $f$ is a non-continuous step function.

**Sign-OPT** Cheng et al. (2020) later improved the query efficiency by only considering the sign of the gradient estimate,

$$\hat{\nabla} g(\boldsymbol{\theta}) \approx \hat{\mathbf{g}} := \sum_{i=1}^{q} \mathrm{sign}\left(g(\boldsymbol{\theta} + \beta \mathbf{u}_i) - g(\boldsymbol{\theta})\right) \mathbf{u}_i. \tag{17}$$

We focus on the Sign-OPT variant, since the findings are more relevant to the current state-of-the-art.

**HopSkipJumpAttack** Similar to Sign-OPT, HopSkipJumpAttack (HSJA) (Chen et al., 2019) uses a zeroth-order sign oracle to improve Boundary Attack (Brendel et al., 2017). HSJA lacks the convergence analysis of Sign-OPT and relies on one-point gradient estimate. Regardless, HSJA is competitive and can excel in the $L_\infty$ setting.

**Dimension-reduced Sign-OPT & HSJA.** In general, for attacks relying on the Cheng et al. (2019) formulation, the update in Equation 16 becomes

$$\hat{\mathbf{g}} = \frac{1}{q} \sum_{i=0}^{q} \frac{g(\boldsymbol{\theta}' + \beta \mathbf{u}_i') - g(\boldsymbol{\theta}')}{\beta} \cdot \mathbf{u}_i' \tag{18}$$

for the reduced-dimension Gaussian vectors $\{\mathbf{u}_i' \in \mathbb{R}^{d'}\}_{i=0}^{q}$ for integer $d' < d$ and direction $\boldsymbol{\theta}' \in \mathbb{R}^{d'}$. The reduced-dimension direction $\boldsymbol{\theta}'$ is initialized randomly with $\boldsymbol{\theta}' \sim \mathcal{N}(0, 1)$ for the untargeted case, or for the targeted case as $\boldsymbol{\theta}' = \mathcal{E}(\mathbf{x}_t)$, where $\mathbf{x}_t$ is a test sample correctly classified as target class $t$ by the victim model. This scheme also applies to HSJA, since HSJA performs a single-point sign estimate. As in the normal variants, $\hat{\mathbf{g}}$ is used to update $\boldsymbol{\theta}'$.

## A.3 MAIN PAPER BLOCK DIAGRAM

A block diagram of assumptions, claims, and observations is shown in Figure 3.

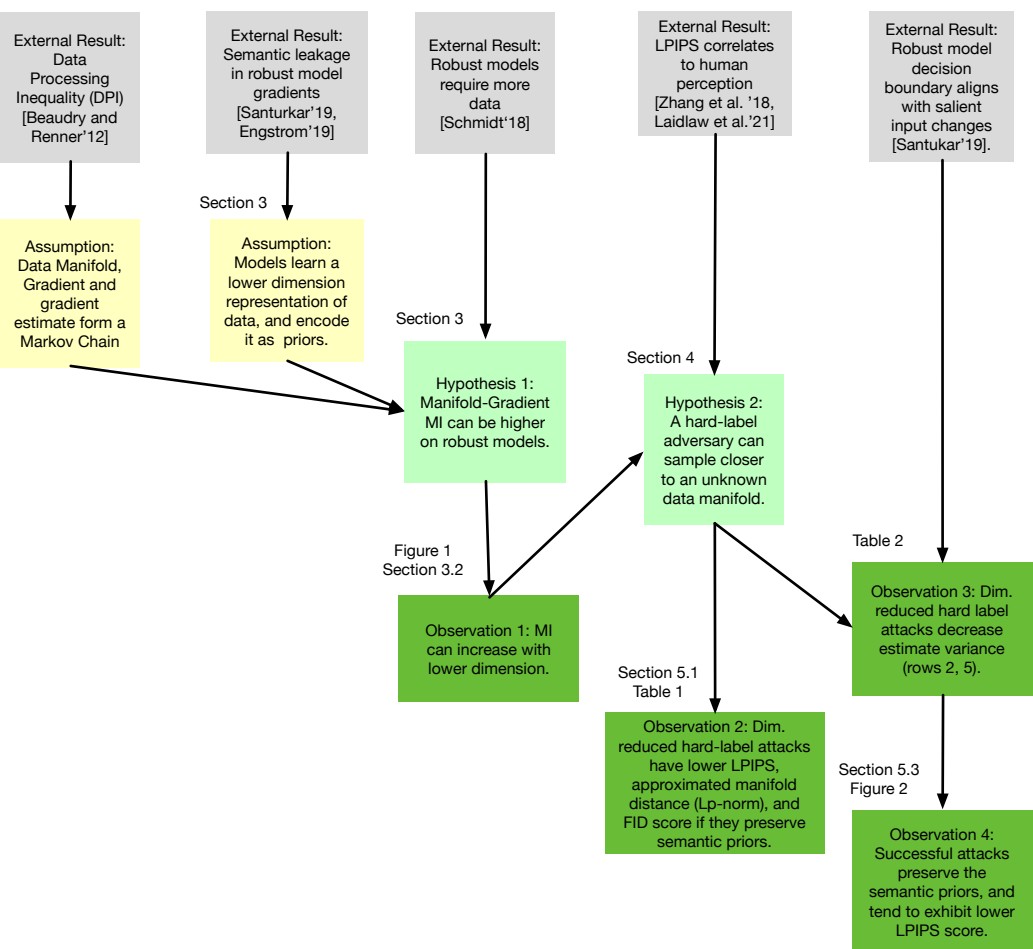

Figure 3: Block diagram summarizing the assumptions, claims, and observations of the main paper.

### A.4 IMPLEMENTATION DETAILS

#### A.4.1 HARDWARE AND ATTACK HYPERPARAMETERS

All experiments in the main paper were performed on an internal high-performance compute cluster equipped with NVIDIA Tesla V100 Tensor Core GPUs and high-speed non-volatile flash storage. In total 16 GPUs, 1TB main system memory, and 40 Intel Xeon CPU cores were used to run experiments completely.

Depending on dataset dimension, HSJA requires tuning of parameter $\gamma$ for best performance. On CIFAR-10 we used $\gamma = 10.0$. For ImageNet, it was necessary to set $\gamma \geq 1000.0$ to re-create the published results of the regular variant (Chen et al., 2019). Due to similar performance we use $\gamma = 1000.0$ for regular and dimension-reduced variants. We note that the dimension-reduced variants like HSJA+BiLN were less sensitive to $\gamma$, performing similarly regardless of the setting.

#### A.4.2 ADVERSARY AUTOENCODER

We are primarily interested in the effect of reduced search resolution on attack behavior. Thus in this work, given a candidate direction $\boldsymbol{\theta}'$ and magnitude (or radius) $r$, the adversarial sample in the AE case is the blending $(1 - r)\mathbf{x}_0 + r\mathcal{D}\left(\mathcal{E}(\mathbf{x}_0) + \boldsymbol{\theta}'\right)$.[3]

For AE attack variants, we implement the same architecture described by Tu et al. (2019). Specifically it leverages a fully convolutional network for the encoder and decoder. Every AE is trained using the held out test set, as we assume disjoint data between adversary and victim.

The adversary's AE is tuned to minimize reconstruction error of input images, so the output quality of the AE will depend on the adversary's ability to collect data. We assume the adversary only has access to the test set, which tends to be considerably less informative than the training set. This crude manifold approximation can manifest as an additional layer of distortion on top of adversarial noise. With BiLN, no additional training is required, so it synthesizes search directions independent of the adversary's manifold description (i.e., possible extracted knowledge about test samples).

ImageNet samples are downsized to 128x128 before passing to the AE, and the output of the AE is scaled back to 224x224, as described by Tu et al. (2019).

#### A.4.3 DATA SAMPLING

Original samples are chosen from the test set using the technique from Chen et al. (2019): on CIFAR-10, five random samples are taken from each of ten uniform-randomly chosen classes (i.e., 50 total samples). On the ImageNet dataset, ten random classes are uniform-randomly chosen and ten random samples taken from each (100 total samples).

### A.5 SUPPLEMENTAL RESULTS

#### A.5.1 QUERY VS. DISTORTION PLOTS

We show the model queries against attack distortion measurement in Figure 4 to accompany the results in the main paper. The distortion is much higher and stays higher with Rand variants, due to discarding important semantic information. The plots evidence that BiLN variants (yellow lines) offer a simple yet effective way to improve the query efficiency of the hard-label attacks.

#### A.5.2 GRADIENT DEVIATION ON ROBUST CIFAR-10

In Table 3 we show supplementary gradient deviation results for CIFAR-10 using different defense mechanisms or robust models. In general they exhibit the same trend as our main paper results, which is that dimension-reduced attacks manage to reduce gradient deviation across each robust model.

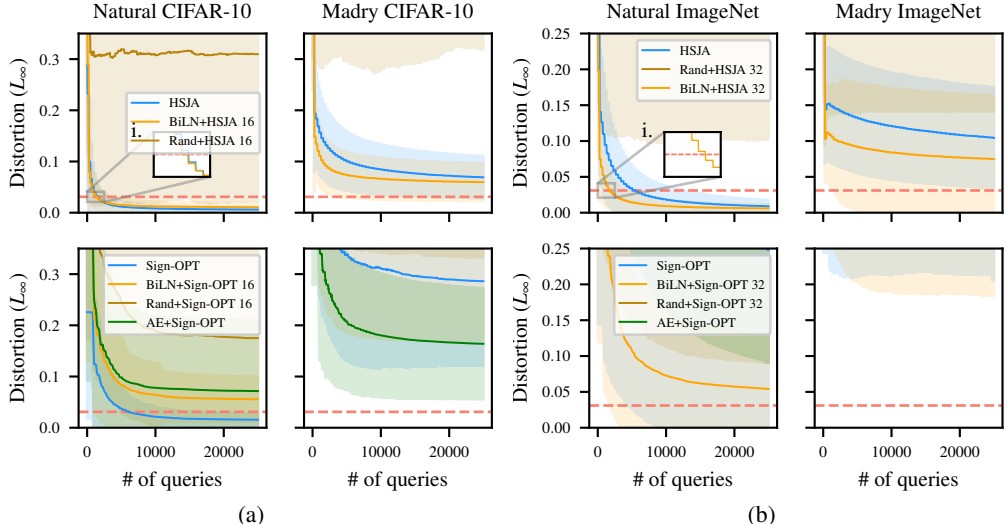

Figure 4: Query vs. distortion plots for a) CIFAR-10 and b) ImageNet, corresponding to the success rate plots in the main text. Dashed lines denote the value of $\epsilon$.

| Attack Variant | TRADES (Zhang et al., 2019) | Interpolation (Zhang & Xu, 2020) | Feat. Scattering (Zhang & Wang, 2019) | SENSE (Jungeum & Wang, 2020) |
|---|---|---|---|---|
| HSJA | 0.0542±0.0001 | 0.0542±0.0001 | 0.0541±0.0000 | 0.0556±0.0045 |
| HSJA+BiLN | 0.0395±0.0001 | 0.0393±0.0001 | 0.0401±0.0004 | 0.0389±0.0056 |
| HSJA+Rand | 0.008±0.004 | 0.002±0.005 | 0.216±0.000 | 0.222±0.017 |
| Sign-OPT | 0.0042±0.0005 | 0.0039±0.0007 | 0.0019±0.0004 | 0.0083±0.0104 |
| Sign-OPT+BiLN | 0.0026±0.0007 | 0.0023±0.0009 | 0.0020±0.0004 | 0.0075±0.0110 |
| Sign-OPT+Rand | 0.007±0.002 | 0.004±0.005 | 0.006±0.002 | 0.025±0.048 |
| Sign-OPT+AE | 0.0257±0.0002 | 0.0282±0.0002 | 0.0259±0.0000 | 0.0278±0.0069 |

Table 3: Per-pixel gradient deviation measured across additional robust CIFAR-10 models

### A.5.3   SUCCESS RATE NORMALIZED AUC SCORES

Tables 5 and 4 show the max-normalized Trapezoid rule area-under-curve (AUC) measurements for the success rate plots of the main text. Highest scores are bolded. Notably, the HSJA+BiLN variant earns the highest score in almost all cases.

### A.5.4   SUCCESS RATE SCORES

We provide the success rates over all samples at specific query intervals in Tables 6 and 7.

### A.5.5   ATTACKING A SMOOTHED MODEL

Gaussian smoothing is a technique of performing adversarial training with sampled affected by Gaussian noise. At test time, inference is achieved via a Monte Carlo search over many Gaussian-perturbed versions of the sample under test. The SotA at time of writing, randomized smoothing proposed by Cohen et al. (2019), is a good candidate for hard-label attacks since the true gradient of the smoothed model is undefined. We use the checkpoint corresponding to smoothing parameter $\sigma = 0.5$ and $\epsilon \simeq 1.0$. These results are shown in Figure 5. In general, the BiLN variant exceeds all other variants in the natural ImageNet case, with small improvement on the smoothed model. Although it can find samples closer to the smoothed $\epsilon$, only a fraction are within the bound.

---

[3]We observed that it is detrimental to set $\mathbf{x} = \mathcal{D}(\mathcal{E}(\mathbf{x}_0) + r\boldsymbol{\theta}')$ or $\mathbf{x} = \mathcal{D}(r\boldsymbol{\theta}')$ directly. Despite remaining on the data manifold by attacking it directly, the approximation of the data manifold is crude, which results in large distortion (Stutz et al., 2019)

| Attack Variant | Natural CIFAR-10 | Madry CIFAR-10 |
|---|---|---|
| HSJA | **1.000** | 0.650 |
| HSJA+BiLN | 0.968 | **1.000** |
| HSJA+Rand | 0.033 | 0.088 |
| Sign-OPT | 0.763 | 0.171 |
| Sign-OPT+BiLN | 0.310 | 0.156 |
| Sign-OPT+Rand | 0.144 | 0.092 |
| Sign-OPT+AE | 0.312 | 0.300 |

Table 4: Success Rate (SR) Normalized AUC scores for CIFAR-10 SR plots of the main text. Higher is better.

| Attack Variant | Natural ImageNet | Madry ImageNet |
|---|---|---|
| HSJA | 0.867 | 0.470 |
| HSJA+BiLN | **1.000** | **1.000** |
| HSJA+Rand | 0.077 | 0.211 |
| Sign-OPT | 0.364 | 0.153 |
| Sign-OPT+BiLN | 0.376 | 0.215 |
| Sign-OPT+Rand | 0.070 | 0.033 |
| Sign-OPT+AE | 0.018 | 0.105 |

Table 5: Success Rate (SR) Normalized AUC scores for ImageNet SR plots of the main text. Higher is better.

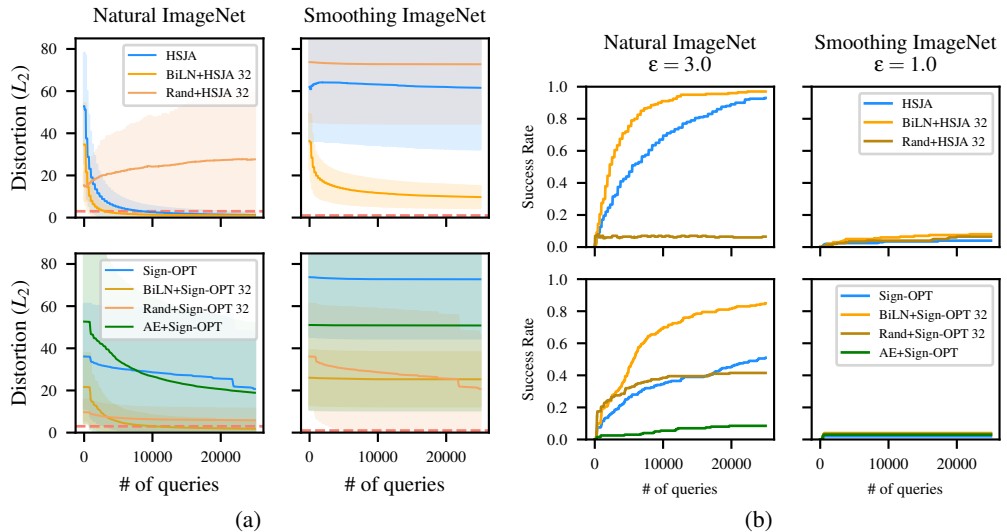

Figure 5: Results of attacking Smoothed ImageNet Cohen et al. (2019) in the $L_2$ setting for a) query vs. distortion and b) query vs. success rate, Dashed lines denote the value of $\epsilon$.

### A.5.6 ATTACKING WITHOUT GRADIENT ESTIMATE

We perform additional experiments with an attack that does not perform an explicit gradient estimate. Chen & Gu (2020) propose an alternative hard-label attack method which is to search for the minimum decision boundary radius $r$ from a sample $\mathbf{x}_0$, along a ray direction $\boldsymbol{\theta}$. Instead of searching over $\mathbb{R}^d$ to minimize $g(\boldsymbol{\theta})$, Chen et al. propose to perform ray search over directions $\boldsymbol{\theta} \in \{-1, 1\}^d$, resulting in $2^d$ maximum possible directions. This reduction of the search resolution enables SotA query

| Attack Variant | Natural @ 4k | Madry @ 4k | Natural @ 11k | Madry @ 11k | Natural @ 25k | Madry @ 25k |
|---|---|---|---|---|---|---|
| HSJA | **0.905** | 0.100 | **0.995** | 0.145 | **1.000** | 0.180 |
| HSJA+BiLN | 0.850 | **0.165** | 0.970 | **0.225** | 0.985 | **0.255** |
| HSJA+Rand | 0.040 | 0.020 | 0.020 | 0.020 | 0.040 | 0.000 |
| Sign-OPT | 0.515 | 0.030 | 0.795 | 0.040 | 0.890 | 0.040 |
| Sign-OPT+BiLN | 0.235 | 0.035 | 0.310 | 0.035 | 0.355 | 0.035 |
| Sign-OPT+Rand | 0.060 | 0.020 | 0.200 | 0.020 | 0.180 | 0.020 |
| Sign-OPT+AE | 0.210 | 0.055 | 0.325 | 0.065 | 0.345 | 0.070 |

Table 6: CIFAR-10 succcess rate values at query intervals 4k, 11k, and 25k, for setting $\epsilon = \frac{8}{255}$..

| Attack Variant | Natural @ 4k | Madry @ 4k | Natural @ 11k | Madry @ 11k | Natural @ 25k | Madry @ 25k |
|---|---|---|---|---|---|---|
| HSJA | 0.550 | 0.105 | 0.850 | 0.130 | 0.965 | 0.165 |
| HSJA+BiLN | **0.835** | **0.240** | **0.965** | **0.290** | **1.000** | **0.335** |
| HSJA+Rand | 0.070 | 0.060 | 0.070 | 0.060 | 0.070 | 0.060 |
| Sign-OPT | 0.210 | 0.045 | 0.335 | 0.045 | 0.485 | 0.045 |
| Sign-OPT+BiLN | 0.240 | 0.055 | 0.345 | 0.065 | 0.445 | 0.065 |
| Sign-OPT+Rand | 0.050 | 0.010 | 0.070 | 0.010 | 0.070 | 0.010 |
| Sign-OPT+AE | 0.015 | 0.030 | 0.015 | 0.030 | 0.020 | 0.040 |

Table 7: ImageNet succcess rate values at query intervals 4k, 11k, and 25k, for setting $\epsilon = \frac{8}{255}$.

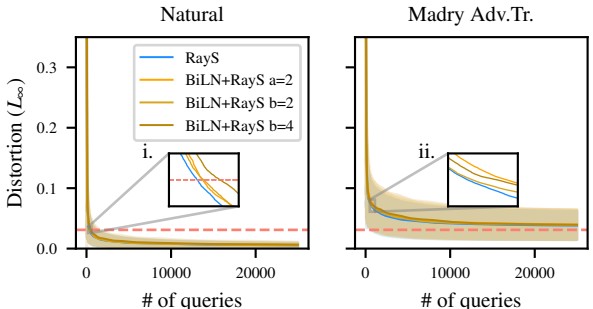

Figure 6: Results for RayS on the CIFAR-10 dataset, corresponding to distortion against query usage (dotted red line denotes the value of $\epsilon$, shaded areas mark standard deviation).

efficiency in the $L_\infty$ setting with proof of convergence. The search resolution is further reduced by the hierarchical variant of RayS, which performs on-the-fly upscaling of image super-pixels.

The intuition behind RayS attack is to perform a discrete search in at most $2^d$ directions. Chen et al. also perform a hierarchical search over progressively larger super-pixels of the image. This has the effect of already upscaling *on-the-fly* (Chen & Gu, 2020). RayS has the unique behavior of performing a discrete search for the decision boundary, rather than an explicit gradient estimate. To achieve an appropriate reduced-dimension version of RayS, we modify the calculation of $s$ in Algorithm 3 of Chen & Gu (2020), which either speeds up upscaling by a factor $a$ (i.e., $s = s + a$), or extends the search through a specific block index by a factor $b$ (increase block level at $k = 2^s b$ instead of $k = 2^s$).

The result of attacking CIFAR-10 with RayS is shown in Figure 6. The BiLN variants of RayS each have minimal effect on overall query efficiency (Insets 6.i and 6.ii). This is a result of RayS not relying on explicit gradient estimation. When comparing the FID-64 score, the dimension-reduced variants of RayS do not have a large variation between them (Inset 6.i), a side-effect of the adaptive super-pixel search, which can automatically scale the super-pixel size as the search progresses.

| | Natural CIFAR-10 | Madry CIFAR-10 | Natural ImageNet | Madry ImageNet |
|---|---|---|---|---|
| Benign | $0.787 \pm 0.830$ | $0.564 \pm 1.724$ | $1.206 \pm 0.803$ | $2.623 \pm 2.383$ |
| HSJA | $8.014 \pm 5.829$ | $62.709 \pm 112.416$ | $4.798 \pm 2.578$ | $3.342 \pm 2.263$ |
| HSJA+BiLN | $7.497 \pm 5.811$ | $50.467 \pm 100.057$ | $4.787 \pm 2.550$ | $4.290 \pm 4.524$ |
| HSJA+Rand | $6.156 \pm 6.053$ | $15.745 \pm 24.195$ | $5.191 \pm 1.948$ | $3.132 \pm 2.489$ |
| Sign-OPT | $7.240 \pm 5.006$ | $51.491 \pm 100.080$ | $4.747 \pm 1.988$ | $3.707 \pm 3.660$ |
| Sign-OPT+BiLN | $6.308 \pm 4.079$ | $47.355 \pm 119.958$ | $4.547 \pm 2.118$ | $4.808 \pm 4.873$ |
| Sign-OPT+Rand | $5.576 \pm 4.178$ | $12.792 \pm 13.546$ | $5.364 \pm 1.757$ | $4.867 \pm 3.369$ |
| Sign-OPT+AE | $6.700 \pm 4.735$ | $51.380 \pm 103.355$ | $4.891 \pm 2.299$ | $3.791 \pm 3.598$ |

Table 8: Measurement of Local Intrinsic Dimensionality (LID) averaged over 200 samples.

### A.5.7 LOCAL INTRINSIC DIMENSIONALITY

In Table 8 we show the average Local Intrinsic Dimensionality Amsaleg et al. (2017) for each dataset and attack combination.

### A.5.8 FRÉCHET INCEPTION DISTANCE

Unfortunately, the data manifold of real-world datasets is difficult to describe. This is an open problem in the study of Generative Adversarial Networks (GANs), since designers require that generator images are on-manifold (i.e., in-distribution Zhang et al. (2020)) to preserve semantic relationships between images. This has motivated the recently proposed Fréchet Inception Distance (FID) that acts as a surrogate measure of the manifold distance over a set of RGB image samples (Heusel et al., 2018). As an additional proxy for manifold distance, we run experiments that assume adversarial samples are synthetically generated images from the data manifold, which can later be compared to their unmodified counterparts on the true manifold using FID. As a result, this estimation process is only available from the defender's perspective. Since FID uses an Inception-V3 coding layer (Szegedy et al., 2016) to encode images, the estimation correlates with distortion of semantic high-level features. Thus sampling closer to the data manifold will result in a lower FID score. The attacks in our experiments do not target the Inception-V3 network, so the FID metric will not rely on any internal aspects of the victim models.

FID score is calculated using the 64-dimensional max pooling layer of the Inception-V3 deep network for coding (denoted as FID-64 in this supplementary material), taken from an open-source implementation.[4] The choice of the 64-dimensional feature layer allows to calculate full-rank FID without the full 2,048 sample count of original FID, which is prohibitive based on the scale of our analysis. Since the coding layer differs slightly from the original FID-2048 implementation, the magnitudes will differ from those published by Heusel et al. (2018).

The comparison of FID scores is shown in Table 9 for natural and robust models. The scores for ImageNet on dimension-reduced attack variants (italicized) are universally lower (as low as 0.014, bold), while on CIFAR-10 the regular variants did not exhibit the behavior. We posit that the higher dimensionality of ImageNet ($224 \times 224$) enables dimension reduction to be more effective than the lower dimension CIFAR-10 ($32 \times 32$). In general, attacks have a higher FID score on robust models than natural models. This can be explained by the fact that robust models are more secure in a region around the original sample, as a result the adversarial sample discovery is further away from the true manifold. The random variant (Rand) in rows three and six evidences that the preservation of semantic priors is important during the update, otherwise samples have high manifold distance. The regular variants of HSJA and Sign-OPT are capable of high FID scores on robust models. However, dimension-reduced variants have a universal behavior to reduce the score in the robust setting, similar to the natural setting for ImageNet. AE variants exhibit higher FID score than BiLN, since BiLN can rescale invariant of the adversary's manifold knowledge (e.g., only having knowledge of test set).

---

[4] https://github.com/mseitzer/pytorch-fid

| Attack Variant | Natural CIFAR-10 | Madry CIFAR-10 | Natural ImageNet | Madry ImageNet |
|---|---|---|---|---|
| HSJA | 0.005 | 1.622 | 1.026 | 29.756 |
| HSJA+BiLN | 0.006↑ | 0.373↓ | 0.012↓ | 4.646↓ |
| HSJA+Rand | 2.198↑ | 8.256↑ | 3.404↑ | 2.354↓ |
| Sign-OPT | 0.001 | 0.305 | 20.969 | 38.505 |
| Sign-OPT+BiLN | 0.002↑ | 0.045↓ | 0.009↓ | 0.062↓ |
| Sign-OPT+Rand | 0.141↑ | 0.210↓ | 0.234↓ | 0.156↓ |
| Sign-OPT+AE | 0.333↑ | 0.008↓ | 1.514↓ | 7.869↓ |

Table 9: Fréchet Inception Distance (FID) scores for each attack's set of 200 adversarial samples on CIFAR-10 and ImageNet (lower is better). $^{\star}$ denotes highest success rate (SR) AUC. Arrows denote higher or lower score compared to baseline variant.

| Attack Variant | Natural CIFAR-10 | Madry CIFAR-10 | Natural ImageNet | Madry ImageNet |
|---|---|---|---|---|
| HSJA | $0.016 \pm 0.012^{\star}$ | $0.162 \pm 0.099$ | $0.030 \pm 0.046$ | $0.170 \pm 0.119$ |
| HSJA+BiLN | $0.033 \pm 0.024$↑ | $0.156 \pm 0.096$↓$^{\star}$ | $0.019 \pm 0.017$↓$^{\star}$ | $0.169 \pm 0.122$↓$^{\star}$ |
| HSJA+Rand | $0.334 \pm 0.176$↑ | $0.457 \pm 0.101$↑ | $0.309 \pm 0.136$↑ | $0.308 \pm 0.141$↑ |
| Sign-OPT | $0.015 \pm 0.013$ | $0.137 \pm 0.088$ | $0.096 \pm 0.118$ | $0.152 \pm 0.112$ |
| Sign-OPT+BiLN | $0.048 \pm 0.039$↑ | $0.191 \pm 0.103$↑ | $0.040 \pm 0.044$↓ | $0.171 \pm 0.105$↑ |
| Sign-OPT+Rand | $0.084 \pm 0.092$↑ | $0.214 \pm 0.100$↑ | $0.082 \pm 0.077$↓ | $0.087 \pm 0.059$↓ |
| Sign-OPT+AE | $0.058 \pm 0.123$↑ | $0.094 \pm 0.068$↓ | $0.235 \pm 0.200$↑ | $0.586 \pm 0.299$↑ |

Table 10: $L_\infty$ distance between adversarial and benign samples projected to approximated manifold (using autoencoder trained on training data) for each attack's set of 200 adversarial samples on CIFAR-10 and ImageNet (lower is better). Arrows denote higher or lower distance compared to baseline variant, and starred items indicate highest success rate.

### A.5.9 $L_\infty$-NORM OVER APPROXIMATE MANIFOLD

We re-use the setup described in Section A.4.2, but train the autoencoders using the training data (defender's perspective) instead of test data (attacker's perspective). The results are shown in Table 10. The HSJA+BiLN attack variants were successful in lowering distance for both natural and robust ImageNet. Generally, Sign-OPT variants were most successful for lowering distance from baseline variant for both CIFAR-10 and ImageNet. The primary factor is the dataset dimensionality, with dimension reduction having a bigger impact on ImageNet than CIFAR-10 (green arrows in ImageNet are more widespread). Likewise, robust models always exhibit a higher distance than natural. This can be explained by the fact that adversarially trained models are more robust in a region around the benign sample, thus the successful adversarial sample will be farther away.

### A.5.10 VISUAL RESULTS - CIFAR-10

We provide visual qualitative results for each attack on CIFAR-10 in Figure 7.

### A.5.11 VISUAL RESULTS - IMAGENET

We provide visual qualitative results for each attack on ImageNet in Figure 8.

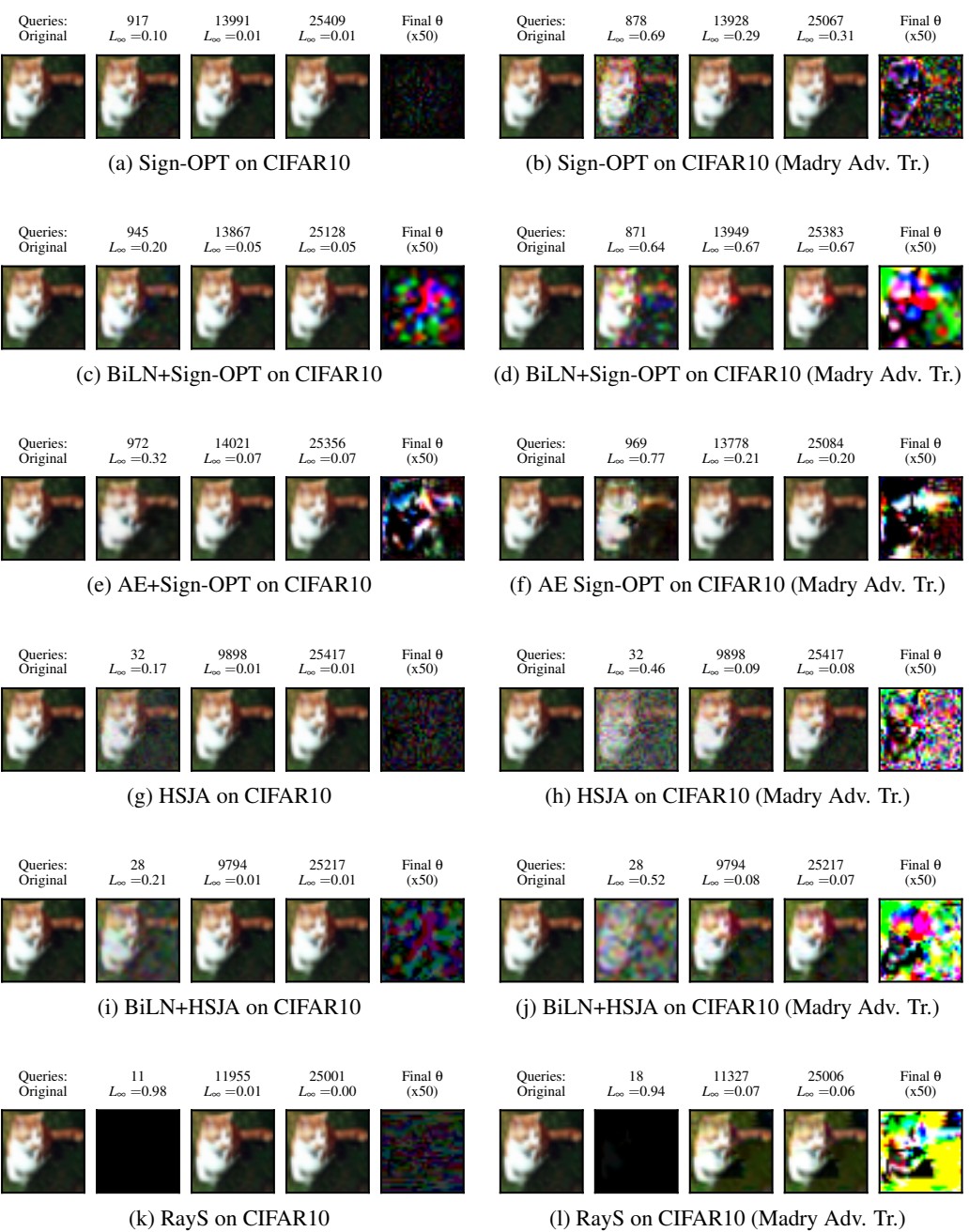

Figure 7: Visual selection of attack trajectories on CIFAR-10.

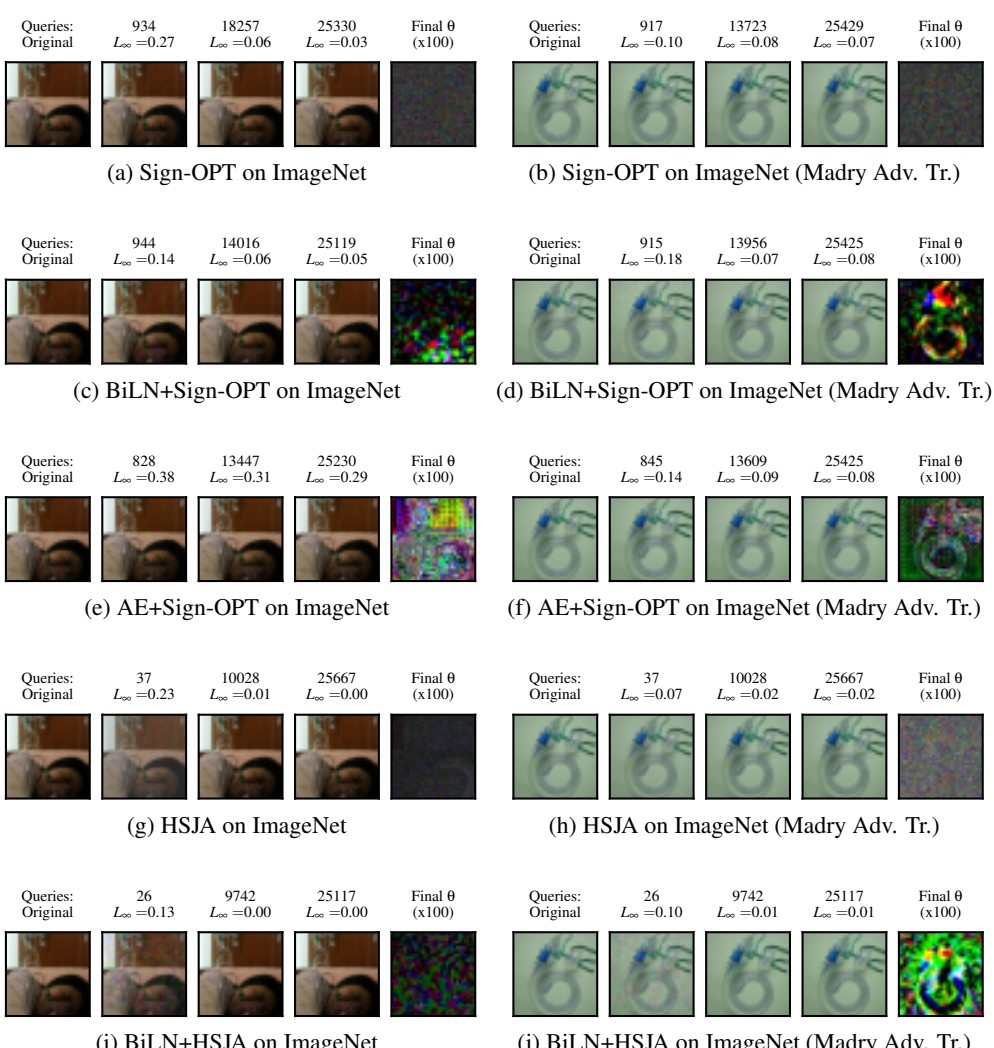

Figure 8: Visual selection of attack trajectories on ImageNet.

