# OpenReview forum: "Less is More: Dimension Reduction Finds On-Manifold Adversarial Examples in Hard-Label Attacks"
_ICLR.cc/2022/Conference — ICLR 2022 Submitted_

### Official Review · Reviewer_iTBK · 2021-10-31

**Correctness:** 3
**Technical Novelty And Significance:** 3
**Empirical Novelty And Significance:** Not applicable
**Recommendation:** 5
**Confidence:** 2

**Main Review:**

[Strength]
1 It is very interesting to see the theoretical illustrations that compared with the natural model, the robust model could provide more information about data manifold (i.e. verification of hypothesis 1), which actually provides theoretical justification of (Santurkar et al. (2019)).

2 This paper further argues dimension-reduced hard-label attacks are more effective on robust models than on standardly trained models.

[Weakness]
1 This paper is based on the assumption that adversarial data are features, not bugs. However, there is evidence that weakens the basic assumption this paper is built on, e.g., adversarial data are just bugs. <https://distill.pub/2019/advex-bugs-discussion/response-5/>

2 Although this paper provides some interesting insights about the hard-label attacks, it does not produce the effective designs of hard-label attack methods, which undermines the paper's significance.

3 The arguments are scattered, which does not form a uniform and compelling viewpoint. Sadly, I still find some experiments in Section 5 cannot well support the paper's arguments.

[Questions]
1 In the top of page 3, the author(s) stated that "The desire for better query efficiency motivated the use of dimension reduction in hard-label attacks. However, to date it is not completely understood how this relates to traversal through the data manifold."
I do not catch up with the logical link of the above statement. Could authors illustrate more to me?

2 I get confused about the linkage between NMD and Hypothesis 1. May I know why NMD definition matters to Hypothesis 1?
Besides, at the bottom of page 5, the authors stated that "the NMD oracle acts as a side channel leaking sensitive information as a factor of the model robustness and data dimensionality."  Could I know what the authors mean about this?  Could I understand in this way that hard-label attacks against a robust model could reveal more sensitive training data information; the low data's dimensionality could worsen this information leakage?
If my understanding were correct, this could contradict the common sense that memoization by standard training could leak sensitive information.

3 In Figure 1, why the variance (shade) is larger for blue lines and smaller for red lines? Besides, it seems that all red, blue, and yellow lines are not well separated.

4 Could I check the success rate of methods stated in Table 1.

5 May I know what is manifold distance. In other words, what is d'' in Hypothesis 2? Could the author(s) provide more details to me?

**Summary Of The Paper:**

[Summary]
This paper is based on a basic assumption that adversarial data are off-manifold and cannot be generated naturally from the true manifold.

This paper provides information-theoretic analysis on the hard-label adversarial attacks from the manifold perspective.


**Summary Of The Review:**

Although this paper provides some interesting points relating to hard-label attacks on robust models, I feel the paper's arguments are scattered, which does not form a uniform and compelling viewpoint. Thereby, I hope the authors could reorganize the paper and highlight a compelling point.



##### Post rebuttal ####
Many thanks for authors' feedback.
I have read other reviewers' comments and corresponding feedback.
I agree with other reviewers' evaluations such as "poor writing", "unclear points", etc.
I keep my score unchanged.

---

> ### Author Response · Authors · 2021-11-19
> **Addressing concerns and related work**
>
> > There is evidence that weakens the basic assumption this paper is built on, e.g., adversarial data are just bugs.
>
> We appreciate the reviewer’s connection to the article. Although the authors describe the creation of adversarial data that are bugs, their construction forces PGD to only step away from the manifold. We propose the opposite, which is to leverage search directions that let an attacker step closer to the manifold. Stutz et al. [1] showed that on-manifold examples exploit the generalization error of the model, or in terms of the article, would yield useful data for training the model (i.e., features).
>
> [1 ] D. Stutz, M. Hein, and B. Schiele, “Disentangling Adversarial Robustness and Generalization,” in IEEE Conference on Computer Vision and Pattern Recognition (CVPR), 2019, p. 12.
>
> > Although this paper provides some interesting insights about the hard-label attacks, it does not produce the effective designs of hard-label attack methods, which undermines the paper's significance.
>
> We offer new insights into hard-label attacker behavior by showing numerically that adversaries can benefit from dimension reduction (Figure 1), which can lead to lower gradient deviation in practice (Table 2) or lower human-aligned distortion (LIPIPS score, Table 1). Although we do not produce an entirely new design, our insights help improve existing designs through a principled application of dimension reduction, evidenced by higher query efficiency in Figure 2 and previously mentioned improvements of LPIPS and gradient deviation. The ICLR guidelines state that not all papers need to propose new methods, in fact new insights are important too. We believe our work falls in the latter category.
>
> > The arguments are scattered, which does not form a uniform and compelling viewpoint.
>
> We provide a block diagram in the Appendix (Figure 3) which may help clarify the flow of arguments.
>
>
> > In the top of page 3, the author(s) stated that "<...>" I do not catch up with the logical link of the above statement.
>
> The convergence speed of zeroth-order optimization algorithms is tied to the gradient estimate dimensionality (Liu et al.), which motivates leveraging a lower-dimension gradient estimate during hard-label attack (such as using an encoder to reduce image dimension, then running the attack on this version). In the process of estimating the gradient, updating the low-resolution image, and then upscaling it, the attacker effectively walks through image space over the course of attack updates. Since the lower dimensionality implies a lower search resolution, we wanted to investigate if the attacker walks closer or farther away to the actual data manifold in the process, due to the information loss. In Figure 1 we get a glimpse that with reduced dimensionality, we gain mutual information between manifold and input gradient, which should yield better search directions in practice. Our empirical results in Tables 1 and 2, and Figure 2 showed that we can enjoy better search directions as a result of reduced dimension. To clarify this point, we have updated the text (red text) in Section 2 to summarize the link between dimension reduction and query efficiency.
>
> > I get confused about the linkage between NMD and Hypothesis 1. May I know why NMD definition matters to Hypothesis 1?
>
> We hypothesized that the NMD oracle can leak more sensitive information with lower dimension, which is supported by our latest numerical results in Figure 1, and the decrease in gradient deviation shown in Table 2 (green arrows) on real data. This can lead to better query efficiency as evidenced by improved success rate in FIgure 2 (and the new Tables 6 and 7).
>
>
> > In Figure 1, why the variance (shade) is larger for blue lines and smaller for red lines? Besides, it seems that all red, blue, and yellow lines are not well separated.
>
> Figure 1 shows that 1) models of varying epsilon behave similarly, and 2) mutual information is higher at lower dimensionality. The lower variance can be explained by the increased sample count for robust cases (due to Schmidt et al. definition). We have updated the figure with the natural model case ($\epsilon=0.0$) and found a similar trend.
>
>
> > Could I check the success rate of methods stated in Table 1.
>
> We have updated the draft to include success rate at different query intervals (Tables 6 and 7 in the Appendix on page 21). The captions are highlighted orange to differentiate the new tables.
>
>
> > May I know what is manifold distance. In other words, what is d'' in Hypothesis 2? Could the author(s) provide more details to me?
>
> Given an encoding back to the true data manifold, the manifold distance is the distance measured between the original sample’s encoding, and the adversarial sample’s encoding. In the hard-label setting, we formulate the noisy manifold distance measure as a binary indicator which tells us if our adversarial sample points towards the manifold (0), or away from it (1).

---

### Official Review · Reviewer_Wy2g · 2021-11-01

**Correctness:** 2
**Technical Novelty And Significance:** 3
**Empirical Novelty And Significance:** 3
**Recommendation:** 5
**Confidence:** 3

**Main Review:**

Strengths:
- The paper is interesting to read and presents several results.
- The topic of understanding adversarial attacks is important and timely.
- The findings on dimension reduction attacks, reduced resolution, and query efficiency seem to be new and possibly significant.

Weaknesses:
- The main shortcoming is the presentation of the paper, which makes it difficult to understand and appreciate. It is often unclear from the text what exactly the authors mean in their arguments and how the arguments are supported.
- The structure of the paper should be improved. Currently, the paper is written as a stream of arguments (definitions, hypotheses, and observations), which are not well supported. For example, it is unclear whether the observations rely only on the presented empirical study.
- In this regard, I believe the experimental study should be more extensive in order to provide a more substantial empirical evidence for the arguments made in the paper.

**Summary Of The Paper:**

The paper addresses hard-label adversarial attacks from a geometric/manifold perspective. Specifically, it examines examples that live close to or on the data manifold. The paper presents a noisy manifold distance based on information-theoretic considerations and proposes three ways to approximate it (taking into account that the data manifold is unknown). Experimental results of HSJA and Sign-OPT attacks on CIFAR-10 and ImageNet are presented.

**Summary Of The Review:**

Interesting paper, but arguments are not sufficiently supported

---

> ### Author Response · Authors · 2021-11-19
> **Updates in new draft**
>
> Thank you for the comments. In response to general reviewer concerns, we have updated the draft to include more extensive experiments which use a higher sample count on non-Rand variants (50->200 samples for CIFAR-10, 100->200 for ImageNet). This was only done on non-Rand variants, since experiments with Rand would not finish before the end of the rebuttal period. To clarify the numerical experiments, we have included the case for $\epsilon=0.0$ and observed that it behaves similarly to the robust case ($\epsilon>0$). We have revised some of the text in response to other reviewer comments, which may help address some of the issues with the structure. The block diagram of arguments in the Appendix (Figure 3) should help to clarify the contribution.

---

### Official Review · Reviewer_cq6A · 2021-11-02

**Correctness:** 1
**Technical Novelty And Significance:** 1
**Empirical Novelty And Significance:** 1
**Recommendation:** 3
**Confidence:** 4

**Main Review:**

The focus of this paper—black-box attacks against publicly available deep learning systems—is an important issue from a security perspective, and thus a key research problem. That being said, the paper is poorly written: without a clear hypothesis, or convincing experimental or theoretical results. Several assumptions and claims are made in the paper without being properly substantiated. I have listed more specific comments/questions below:

[Section 3]

- The motivation behind NMD is not provided. In particular, even if the gradients of the robust model are aligned with the data manifold (which is not a direct takeaway from Santurkar et al.), it is not clear why the adversary would have access to a robust model in the first place.
- The assumptions after Definition 3 are not sufficiently justified. References are cited, however these claims are not made by prior work.
- The upped bound being larger does not imply that I(M, $\"{G}$) is larger.

[Section 4]

- What is the motivation for random BiLN?
- How exactly is the gradient deviation measured? In particular, how (and when) is the true gradient calculated?

[Section 5]

- It is not clear why the LPIPS distance or the gradient deviation are quantities we should care about in practice. After all, in this setting, the adversary would care only about the attack success rate. And as the authors themselves report, there is no direct correlation between LPIPS/gradient deviation and SR.
- The results reported in Tables 1 and 2 do not show a consistent trend from the proposed modifications.
- Results are reported over 50-100 examples of the dataset (which for ImageNet does not even have 1 example per class), and thus could have large statistical fluctuations. Why did the authors choose to report metrics on such small subsets of the data?
- The connections between the theoretical section (which explores the relationship between manifold-gradient MI and dimensionality for robust models), and doing dimensionality reduction in practice for *standard* (non-robust) models isn't provided.

**Summary Of The Paper:**

This paper studies zeroth-order hard-label adversarial attacks. In particular, the authors explore the connection between how the gradient can reveal information about the data manifold, and its dependence on data dimensionality. They also empirically consider attacks with reduced dimensionality in practice.

**Summary Of The Review:**

The presentation of this paper could be substantially improved---including outlining the central hypothesis and performing experimental evaluations in more standard settings.

### Post-rebuttal update

I thank the authors for their detailed response. Unfortunately, my concerns with the paper regarding the presentation and motivation (which are also shared by other reviewers), as well as the practical significance of the proposed method still hold. In particular:

- I believe that the paper would benefit from a significant rewrite to clarify precisely the main claims, justify assumptions, and better connect the different sections.

- Regarding the experimental findings, the canonical measure of success of black-box-attacks is the ASR for a fixed epsilon. In this regard, does not seem to be a significant improvement. Moreover, the statistical significance of the findings is hard to verify---in Tables 1 and 2, the differences w.r.t. baselines often lie within confidence intervals, and in many cases the standard deviation is larger than the mean itself. The time constraints of the rebuttal phase are not a justification for not providing results over enough samples. The authors should have verified the statistical significance of their results (by increasing the number of samples till they get reasonable confidence intervals) as part of the original submission.

---

> ### Author Response · Authors · 2021-11-19
> **Addressing concerns with motivation**
>
> > The motivation behind NMD is not provided. In particular, even if the gradients of the robust model are aligned with the data manifold (which is not a direct takeaway from Santurkar et al.), it is not clear why the adversary would have access to a robust model in the first place.
>
> Adversarial ML is considered a natural way to expose the worst-case robustness of ML models; but throughout our numerical and empirical analysis, we show that using a robust model (compared to natural model) will actually provide a similar or better position to the adversary, as measured by gradient deviation (Table 2). When leveraging dimension reduction, an adversary can improve their search directions, as we show with LPIPS scores (Table 1) which are indeed smaller. The hard label setting itself is agnostic to whether a model is robust or not. The NMD oracle is beneficial in this setting since the adversary cannot directly query the model’s manifold encoding, only the decision function.
> In regards to the motivation of manifold alignment from Santurkar et al. (2019), we believe this is a direct take-away due to the following from Section 2 of their paper. For context, Santurkar et al. discuss in a preceding paragraph that under robust training, there is a phenomena where changes in the model’s predictions correspond to salient input changes (investigated by Tsipras et al. 2019):
>
> “This indicates that robust models exhibit more human-aligned gradients, and, more importantly, that we can precisely control features in the input just by performing gradient descent on the model output.”
>
> Since gradients of robust models are human-aligned, they can be said to align with a low-dimensional representation of the data. Our motivation is likewise substantiated by Tsipras et al (2019), who show that salient input changes (i.e., traversals on the data manifold) correspond to changes in robust model prediction.
>
>
> > The assumptions after Definition 3 are not sufficiently justified. References are cited, however these claims are not made by prior work.
>
> The references give examples of previous encoding functions which map raw image space (pixels) to lower-dimensional representations that encode the image manifold. Stutz et al. train encoders explicitly, whereas Zhang et al. use the output of intermediate convolutional neural network layers as encoders. The distance function can be replaced by a measure that reliably indicates perceptual distortion (as proposed by Zhang et al. with LPIPS). If the reviewer is referencing a different definition, we would be glad to clarify.
>
>
> > What is the motivation for random BiLN?
>
> Our down-sampling methods preserve semantic information of the image because they do not scramble the spatial information. Random BiLN was implemented as a baseline downsampling method that effectively scrambles the spatial dimensions while performing the dimension reduction.
>
>
> > How exactly is the gradient deviation measured? In particular, how (and when) is the true gradient calculated?
>
> For a successfully classified image, we generate a successful untargeted adversarial sample using an attack. When the adversarial sample is fed to the victim model, it produces an incorrect label.. The true gradient is calculated (with respect to the original sample) using the original criterion of the model and the adversarial (incorrect) label. In our paper, all models used cross-entropy loss for training. We have updated the draft to include these details in Section 4.2 (red text). To clarify, this true gradient (corresponding to first update) is compared against the first step of hard-label attack.
>
>
> > It is not clear why the LPIPS distance or the gradient deviation are quantities we should care about in practice. After all, in this setting, the adversary would care only about the attack success rate. And as the authors themselves report, there is no direct correlation between LPIPS/gradient deviation and SR.
>
> Zhang et al. show through human studies that LPIPS score is a reliable measure of an image sample’s distortion compared to the original. As a result, an adversary should lower LPIPS distance in order to reduce the human-visible distortion of the image. Although our dimension-reduced attacks rely on $L_\infty$-norm to measure success rate, we show they also reduce the LPIPS score compared to the original attacks. The gradient estimation used in hard-label attacks leads to an error, so reducing this error (i.e., gradient deviation) is generally beneficial for increasing query efficiency and reducing image distortion. Although we show some notable exceptions, it is still standard in the literature to lower gradient deviation for better attack convergence.

---

> ### Author Response · Authors · 2021-11-19
> **Clarifying evaluation**
>
> We clarify additional issues with the evaluation, particularly reporting on new results in the updated draft.
>
> > The results reported in Tables 1 and 2 do not show a consistent trend from the proposed modifications.
>
> Our proposed dimension reduction has lower LPIPS and per-pixel gradient deviation for single-point HSJA as the problem dimensionality increases (increasing from Robust CIFAR to Natural ImageNet, and again to Robust ImageNet).
>
> > Results are reported over 50-100 examples of the dataset (which for ImageNet does not even have 1 example per class), and thus could have large statistical fluctuations. Why did the authors choose to report metrics on such small subsets of the data?
>
> We follow the methodology of previous hard-label papers which use between 50 and 100 samples depending on the dataset. To address the reviewer’s concern, we updated the draft with augmented results that use 200 samples (both CIFAR-10 and ImageNet) for non-Rand attacks. Due to the time constraints of the rebuttal phase, we cannot augment Rand variants since they are the longest attack experiment, requiring on the order of 2 weeks wall-time to augment to 200. Thus these variants are kept at 50 samples for CIFAR-10, and 100 for ImageNet. As before, any statistical fluctuations are shown in tables ($\pm$) or as shaded areas in query-distortion graphs (Figure 4 in the Appendix).
>
> > The connections between the theoretical section (which explores the relationship between manifold-gradient MI and dimensionality for robust models), and doing dimensionality reduction in practice for standard (non-robust) models isn't provided.
>
> We have updated Figure 1 in the draft with numerical results for the standard model case ($\epsilon=0.0$) along with a wider sweep of $d \in [5, 2000)$ (The blue line in Figure 1 now represents the natural case, and red text was added to describe it in Section 3.2). In this simple setting, the new result shows that robust models in fact leak as much information for low $d$ as natural models.

---

> ### Author Response · Authors · 2021-11-23
> **Follow-up regarding the mutual information upper bound**
>
> We would like to clarify the following concern about the mutual information upper bound:
>
> > The upper bound being larger does not imply that $I(M, \ddot{G})$ is larger.
>
> Our use of DPI does not guarantee that $I(M, \ddot{G})$ will be larger, it only suggests a connection that may be of interest to the adversary. Our experiments measuring per-pixel gradient deviation (Table 2) help substantiate this, in summary showing that gradient deviation is either universally lower (CIFAR-10) or similar (ImageNet) on robust models.

---

### Official Review · Reviewer_tKPt · 2021-11-02

**Correctness:** 3
**Technical Novelty And Significance:** 2
**Empirical Novelty And Significance:** 3
**Recommendation:** 5
**Confidence:** 2

**Main Review:**

I am not completely familiar with this line of research, and thus cannot properly position the current work w.r.t. existing literature. Therefore, I have lowered my confidence until I see other reviewers' comments. But, for now:

Paper is hard to read, unless the reader has complete and exact knowledge about at least 2 or 3 previous works. Lots of technical terms, specific algorithms and methodologies have been borrowed from the said works, without any discussion or re-definition inside the main body. IMO, this has made the paper less informative. For example, it might not be a bad idea to briefly overview zeroth-order hard-label attacks, or "boundary-tilting assumption" which according to author(s) plays an important role in this paper. In particular, this part:
- "We investigate the scenario where an adversary uses zeroth-order information to estimate the desired gradient direction (Cheng et al., 2020; Chen et al., 2019). Thus the adversary uses only the top-1 label feedback from their model query to synthesize samples. The desire
for better query efficiency motivated the use of dimension reduction in hard-label attacks."

of the "Related Works" section can be replaced by one or two paragraphs that briefly describe the main approaches behind each of these sentences. Another example is that author(s) have repeatedly referred to a new MI measure which 1) helps the adversary to craft on-manifold attacks, and also 2) its value "increases as a function of data dimensionality". However, the meaning and possible significance of the latter argument is vague and non-informative, at least for a non-familiar reader.

Author(s) have relied on a claimed argument, that MI between manifold distribution and the gradient of a robust model is larger than that of an ordinary model, in order to propose their own method of leveraging manifold information during the attack. Is it rigorously justified with exact theoretical arguments? or the claim about MI value only supports the proposed method, intuitively? In other words, do you have any "theorems" to theoretically support your method? If the answer is no, then this paper is more of an experimental investigation of an idea, rather than a theoretical analysis.

Moreover, the transitions from some parts of the paper to the next parts are not smooth. For example, transition from Hypothesis 1 to Definition 3.4 can be made more smooth through adding proper explanations. Definition 3.6 is vague. Please add some explanations on how the given formulation corresponds to the MI between low-dimensional manifold distribution $\mathcal{M}$ and the gradient distribution (?) $\mathcal{G}$. Let us begin with what is $p_{\mathcal{G}}(1)$?

I haven't checked the experimental parts.

----------------------------------------------------------------

Minor comments and suggestions:
- Please correct the citation format at the end of Par. 1 in Introduction section.

**Summary Of The Paper:**

This paper concerns zeroth-order hard-label adversarial attacks on machine learning models, and in particular, how to strengthen them through leveraging manifold information. The core idea is inspired from a series of intuitions and observations around the fact that the gradient of a robust model has more information regarding the underlying low-dimensional manifold of the data, compared to that of a naturally trained model. Author(s) have built upon this inspiration and proposed the noisy manifold distance oracle, which can leak manifold information to the adversary during forging its attack. Author(s) have also implemented their method on a number of real-world datasets.

**Summary Of The Review:**

I am not completely familiar with this line of research, and thus cannot properly assess the current work. IMO, paper suffers from non-informative and cryptic explanations. This issue can be solved by briefly discussing the many technical terms and concepts that paper has implicitly or explicitly utilized.

Also, it seems that this work is not completely grounded on a firm theoretical ground. A number of core ideas and intuitions have been triggered by some theoretical analysis, but have been mostly investigated through experiments.

At this moment, my vote is weak reject (with a low confidence). However, I want to see other reviewers' comments and study a number of references inside the manuscript to have a better judgement.

---

> ### Author Response · Authors · 2021-11-19
> **Addressing concerns**
>
> > Paper is hard to read, unless the reader has complete and exact knowledge about at least 2 or 3 previous works
>
> We have revised the text in Section 2 (denoted with red text) to briefly overview the ideas behind hard-label attack formulation (based on the random gradient-free method), and introduce the difference between the two attacks (two-point and single-point formulation based on search function gradient). We welcome any further comments regarding this section.
>
> > Is it rigorously justified with exact theoretical arguments? or the claim about MI value only supports the proposed method, intuitively?
>
> We empirically verify our method by deriving a closed-form expression (in integral form) under the Gaussian mixture model of Shmidt et al. (2018). We run simulations to compute the MI, and have added results for the standard model case ($\epsilon=0.0$) in the updated draft along with a wider sweep of $d \in [5, 2000)$ (The blue line in Figure 1 now represents the natural case, and red text was added to describe it in Section 3.2). We multiply each case in Equation 1 by a large constant ($10^4$). In this simple setting, the new result shows that robust models in fact leak as much information for low $d$ as natural models.
>
> > Please add some explanations on how the given formulation corresponds to the MI between low-dimensional manifold distribution M and the gradient distribution (?) G. Let us begin with what is P_G(1)?
>
> In our Gaussian mixture model, the input gradient distribution is defined as a Rademacher distribution (due to sign classifier from Schmidt et al. data model), so input gradients take a value of either -1 or 1. P_G(1) should be interpreted as being $P(g_k=1)$, or in other words the marginal probability that gradient equals 1 at dimension $k$.

---

### Author Response · Authors · 2021-11-19
**Summarizing latest changes**

We thank the reviewers for their careful feedback, and attempt to address general concerns found with the paper by summarizing key changes in the new draft.

* A common concern was the lack of numerical experiments for the natural model case in Figure 1. We now include results for epsilon values in $0.0, 0.180$ and $0.25$, so that the blue line now represents the natural case, and added supporting text for this new experiment in Section 3.1 (highlighted with red text). We observe that robust models in fact leak as much information for low $d$ as natural models in terms of mutual information.
* We have augmented the attack sample count in our empirical results to address concerns about statistical fluctuation, increasing from 50 to 200 for CIFAR-10 and 100 to 200 for ImageNet. Due to time constraints of the rebuttal phase, we can only augment the non-Rand attack variants. The relevant tables and figures in the main paper reflect the augmented sample count. There was effectively no deviation from the original results, and our observations remain the same.
* The overall problem motivation and supporting references were clarified with new text in Section 2. This is highlighted in red and tries to briefly summarize the motivation for dimension-reduction, the general idea of hard-label attacks, and a brief introduction to Sign-OPT and HSJA attack formulation.

---

### Author Response · Authors · 2021-11-23
**Follow-up for conclusion of second discussion stage**

Today is the last day to allow paper updates before the final discussion stage begins. We welcome any feedback or suggestions that would ease concerns with the current revision, and look forward to discussion in the final stage.

---

> ### Author Response · Authors · 2021-11-29
> **Follow-up message from authors**
>
> Dear Area Chair and Reviewers,
>
> We thank Reviewer cq6A for updating the post-rebuttal comments. As the discussion deadline is closing soon, we would like to follow up to ensure we have successfully conveyed the merits and main contributions of our work. We took the silence of the post-rebuttal discussion as a positive sign indicating our revised version and responses had addressed your concerns. In the meantime, we are happy to answer any questions the AC and Reviewers may have. Please don't hesitate to let us know!
>
> Sincerely,
>
> Authors

---

### Decision · Program_Chairs · 2022-01-20

**Decision:**

Reject

**Comment:**

In this work, authors study query efficiency in the zeroth-order setting of adversarial examples. Reviewers pointed out several weaknesses in the work. They mentioned the paper is not well-organized and poorly written, experiments are not comprehensive and the practical significance of the proposed method is unclear. Although reviewers appreciated authors' efforts and responses in the discussion period, they felt that the paper is not above the accept threshold this round and still needs a bit more work.